# Partial response electromyography as a marker of action stopping

**Liisa Raud[1,2]\*, Christina Thunberg[2,3], René J Huster[2,3]**

[1]Center for Lifespan Changes in Brain and Cognition, Department of Psychology, University of Oslo, Oslo, Norway; [2]Cognitive and Translational Neuroscience Cluster, Department of Psychology, University of Oslo, Oslo, Norway; [3]Multimodal Imaging and Cognitive Control Lab, Department of Psychology, University of Oslo, Oslo, Norway

**Abstract** Response inhibition is among the core constructs of cognitive control. It is notoriously difficult to quantify from overt behavior, since the outcome of successful inhibition is the lack of a behavioral response. Currently, the most common measure of action stopping, and by proxy response inhibition, is the model-based stop signal reaction time (SSRT) derived from the stop signal task. Recently, partial response electromyography (prEMG) has been introduced as a complementary physiological measure to capture individual stopping latencies. PrEMG refers to muscle activity initiated by the go signal that plummets after the stop signal before its accumulation to a full response. Whereas neither the SSRT nor the prEMG is an unambiguous marker for neural processes underlying response inhibition, our analysis indicates that the prEMG peak latency is better suited to investigate brain mechanisms of action stopping. This study is a methodological resource with a comprehensive overview of the psychometric properties of the prEMG in a stop signal task, and further provides practical tips for data collection and analysis.

## Editor's evaluation

The authors propose that the covert latency of the stopping process, which allows for the stopping of movements in tasks like the stop-signal paradigm, can be measured through the offset latency of bursts of electromyographic (EMG) activity that is observable on trials in which no overt response (typically a button press) was produced. The investigation is extensive and systematic and will provide a helpful methodological resource that rigorously specifies this alternative measure of stopping.

**\*For correspondence:**
liisa.raud@psykologi.uio.no

**Competing interest:** The authors declare that no competing interests exist.

## Introduction

Response inhibition, or the ability to stop actions, is a core construct of cognitive control and is widely studied among cognitive and clinical neuroscientists. Most of our knowledge about inhibitory capability is derived from a single behavioral measure, the stop signal reaction time (SSRT), which quantifies the time the stopping process needs to countermand an already initiated action. SSRTs are slower in patients with attention deficit/hyperactivity disorder, obsessive compulsive disorder, drug abuse, schizophrenia, Parkinson's disease, and Tourette syndrome among others (*Jahanshahi et al., 2015*; *Lipszyc and Schachar, 2010*; *Smith et al., 2014*; *Snyder et al., 2015*; *Wylie et al., 2013*; *Yaniv et al., 2017*). However, accumulating evidence suggests that SSRT estimations may be unreliable in many practical settings (*Bissett et al., 2021b*; *Bissett and Logan, 2014*; *Matzke et al., 2017*; *Skippen et al., 2019*; *Verbruggen et al., 2013*). Here, we scrutinize electromyography (EMG) recordings from

the muscles that are successfully stopped as a potential physiological marker to capture individual differences in action stopping.

The stop signal task is among the most common experimental tasks used to study inhibition, with more than 7000 publications using the task in the last years (*Verbruggen et al., 2019*). The principle of the task is very simple: participants react repeatedly to a go signal with a button press but need to inhibit this response when a stop signal appears after the go signal with a variable delay (stop signal delay, SSD). The resulting go reaction time distribution, probability of responding despite stop signal presentation, and the resulting SSDs are the necessary variables for calculating the SSRT.

The SSRT estimation is based on the independent race model (or horse race model), in which go and stop processes independently race against each other, and whichever process wins this race, determines whether a planned action will be executed or not (*Band et al., 2003*; *Logan and Cowan, 1984*). The theoretical basis of the model is well established and an extension of this model also has biological plausibility (*Boucher et al., 2007*). However, recent work indicates that SSRT estimates may be inaccurate in many practical cases, for example due to extensive slowing of go responses (*Verbruggen et al., 2013*), violations of the independence assumptions at short SSDs (*Bissett et al., 2021b*), or failures to initiate the stop process (*Skippen et al., 2019*). The available solutions focus on the derivation of putatively improved SSRT estimates, either by refining the estimation method (*Verbruggen et al., 2019*), discarding participants or trials that show obvious violations of the model assumptions (*Bissett et al., 2021a*; *Bissett et al., 2021b*), or by proposing alternative modelling approaches (*Heathcote et al., 2019*; *Matzke et al., 2017*; *Matzke et al., 2013*). However, the field has been in a remarkable standstill when it comes to the development of complementary measures that could potentially bypass issues related to model assumptions in the first place.

A notable exception is the proposition to derive a more direct measure of the stopping latency from EMG. Specifically, EMG recorded at the effector muscles that generate the response can be used to capture the time-point when the motor activity begins to decline in the muscles. In many successfully stopped trials (trials without a registered response), surface EMG recordings still capture small bursts of muscle activity. These bursts are presumably responses initiated after the go signal that are stopped before the muscle activity culminates in a button press. We have termed these responses partial response EMG (prEMG; see *Box 1* for visualization and reflections on the naming). The proposition to use the peak latency of the prEMG as an indicator of an individual's stopping speed seems to have originated independently in two different laboratories (*Atsma et al., 2018*; *Raud and Huster, 2017*). Raud and Huster used surface EMG from thumb muscles in a selective stop signal task and measured the prEMG peak latency to be about 150 ms, which was significantly shorter than the corresponding SSRT of about 200 ms. Atsma et al. used intramuscular recordings of the upper-limb muscle during reaching movements. They reported a prEMG offset latency (the time-point when muscle activity returned to the baseline level) at 165 ms, compared with an SSRT of 244 ms. While their methods and goals were different, both studies showed that the EMG-derived stopping latency was much faster than that of the model-based SSRT.

While using the prEMG latency as an index of stopping capability is a relatively recent suggestion, recording EMG in a stop signal task is not particularly novel by itself. Earlier work identified that muscle activity can be stopped at any time before the movement is completed, and additionally focused on the time-courses of different electrophysiological measures, that is EMG, electrocardiography, and electroencephalography (EEG; *de Jong et al., 1990*; *Jennings et al., 1992*; *van Boxtel et al., 2001*). These studies identified the motor cortex as a site of inhibition, yet they showed that the peripheral nervous system is affected as well. Several studies have attempted to interpret the prEMG within the context of the horse race model. For instance, trials with prEMG seem to roughly correspond to the middle part of the go-RT distribution, indicating that these trials have a relatively fast go process that nonetheless is slow enough to be suppressed by the stop process (*de Jong et al., 1990*; *van Boxtel et al., 2001*). Furthermore, the occurrence of the prEMG is dependent on the SSD, as prEMG frequency declines and its amplitude increases with increasing SSD (*Coxon et al., 2006*). Interestingly, EMG in unsuccessful stop trials may be affected by the stop process as well, as these trials exhibit lower EMG amplitudes, but prolonged motor times (the time between EMG onset and the registered response) compared with standard go responses (*van de Laar et al., 2014*). This violates the assumption of independence underlying the independent horse race model, which states that unsuccessful stop trials are drawn from the distribution of fast go responses and should therefore

## Box 1. Definition and naming of the partial response EMG (prEMG).

prEMG refers to small muscle bursts in successful stop trials (third panel in *Box 1—Figure 1*). These bursts are elicited in response to the go signal but stopped in response to the stop signal before the muscle activation culminates in a button press. As such, the trial itself is considered a successful stop trial, but the prEMG onset and peak latencies indicate the exact timing of the go- and stop processes at the effector muscles, respectively.

Whereas we use the term prEMG here, this term has not been used consistently across studies. However, we believe prEMG to describe the phenomenon most accurately without any assumptions about the underlying mechanisms. Below, we list the terms found in the previous literature.

**Partial EMG**. Early papers used this term and it is a more concise version of the term prEMG. However, partial EMG implies that the EMG signal itself is incomplete, which is not the case. Rather, the EMG signal in successful stop trials itself is perfectly valid but different from other EMG bursts because the response is not completed.

**Interrupted EMG**. *McGarry and Franks, 2003*; *McGarry and Franks, 1997* used this term for muscle bursts in stop trials that had a similar shape at their onset to the go EMG, indicating a slow stopping process after the motor pool activation by the go process. They further dissociated *interrupted response EMG* from *partial response EMG*, the latter having accumulation rates and milder slopes at their onset. This was interpreted as a weakened activation of the motor pool (or a leakage signal) in trials where go EMG was interrupted early, before reaching the motor pool. However, they used a stop signal task with unusually short SSDs from −100 to 100 ms relative to the go signal. Dissociation between these response types is unlikely in a more typical stop signal task with SSDs >100 ms.

**Partial error**. This term is consistently used in tasks in which an erroneous automatic response needs to be overridden by a correct response (e.g., *Burle et al., 2002*; *Coles et al., 1995*; *Rochet et al., 2014*). In such tasks, the term 'error' is appropriate. However, the prEMG in the stop signal task is not erroneous as it is elicited by the go signal, which is perfectly in line with the task instructions.

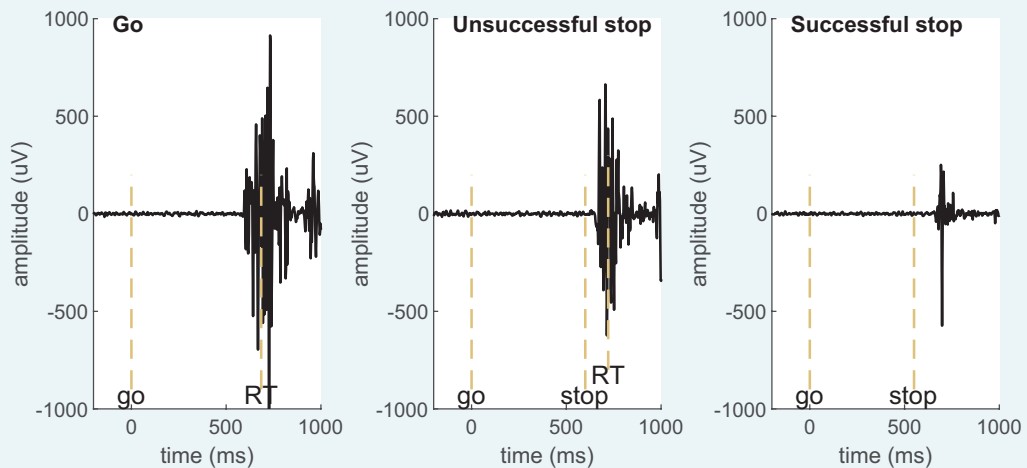

**Box 1—figure 1.** Examples of electromyography (EMG) signals in a go, unsuccessful stop, and successful stop trial in an example participant. EMG in go and unsuccessful stop trials is associated with a button press (marked by RT), while no button press is registered in successful stop trials and the muscle burst therefore qualifies as partial response EMG (prEMG). The dashed vertical lines mark the onset of go signal, stop signal, and the recorded reaction time (RT).

**Subthreshold EMG**. This term was used by *Raud and Huster, 2017*, indicating that the EMG in successful stop trials did not reach the threshold to trigger the button press. However, such thresholds are relatively arbitrary and depend on the physical properties of the response device. In addition, this naming becomes inappropriate if continuous movements are of interest, as for example is the case with measurements of force or movement trajectories. **Cancel time**. This term, suggested by *Jana et al., 2020*, specifically refers to the prEMG peak latency. Whereas appealing due to its simplicity, it is potentially misleading as it implies a direct cancellation mechanism for stopped movements. Yet, alternative mechanisms may cause the same EMG phenomenon, such as the activation of antagonist muscle groups, or a simple reduction in response maintenance.

be unaffected by the stop signal. The dependence of go and stop processes was recently directly shown by *Raud et al., 2020a* who found a positive correlation between go-locked prEMG onsets and stop-locked prEMG peak latencies at the level of single stop trials.

The earlier timing of the prEMG peak latency compared to the SSRT has been replicated in a number of independent studies (*Hannah et al., 2020*; *Jana et al., 2020*; *Raud et al., 2020a*; *Raud et al., 2020b*; *Sundby et al., 2021*; *Tatz et al., 2021*; *Thunberg et al., 2020*). However, while the existence of the prEMG is relatively well established, its characteristics and utility have not been explored sufficiently. We therefore present a thorough investigation of the prEMG in the stop signal task. Importantly, we treat prEMG as a marker of action stopping only, without specifically linking it to inhibitory brain mechanisms, as mechanisms other than active inhibition may cause cessation in muscle activity too. This separation between action stopping and response inhibition holds for the SSRT as well, but is often overlooked in the literature, in which the stop runner in the horse race model is frequently interpreted as an active process to counteract the go process. Nonetheless, as the SSRT and the prEMG have been previously associated with the same underlying neural mechanisms, we benchmark the prEMG peak latency against the SSRT and the existing horse race model framework; investigate potential relationships between response onset, stopping, and the SSD at a single-trial level; present novel information about stopping variability; and estimate the reliability of the EMG measures. We additionally provide practical tips for data collection and analysis, accompanied by publicly available analysis code that can easily be adjusted to new datasets. With that, we demonstrate the full potential of prEMG with the aim to encourage and accelerate future studies to adopt it as a standard outcome measure of supposed inhibition tasks.

## Results

### Behavior and EMG in a stop signal task

We analyzed surface EMG recorded over the thumb muscles (*abductor pollicis brevis*) from 46 healthy young adults who performed a standard stop signal task. The primary task was a choice reaction time task where participants had to respond to a go signal either with a left or right hand button press (*Figure 1A*). In 24% of the trials, a stop signal occurred instructing the participants to stop their response. The full analysis pipeline was additionally applied to a publicly available dataset provided by *Jana et al., 2020* to replicate the current results. The main difference between these two datasets was the proportion of trials with prEMG of successful stop trials, which was relatively low (~20%) in the discovery, but higher (~60%) in the replication dataset. With a few minor exceptions, the results were the same in both datasets. Therefore, we report the results obtained on the discovery dataset, and only report the replication results when these deviate from the discovery dataset results. The complete replication results are available in Appendix 1.

The behavioral results (*Table 1*) were in line with previous studies, with stopping accuracies at 50% and an SSRT (integration method with RT replacement for go omissions) of 199 ms. We also estimated behavioral parameters using the Bayesian Ex-Gaussian Estimation of Stop Signal Distribution approach accounting for go and trigger failures (BEESTS; *Matzke et al., 2019*; *Matzke et al., 2013*), which resulted in an SSRT of 202 ms (*Table 1*).

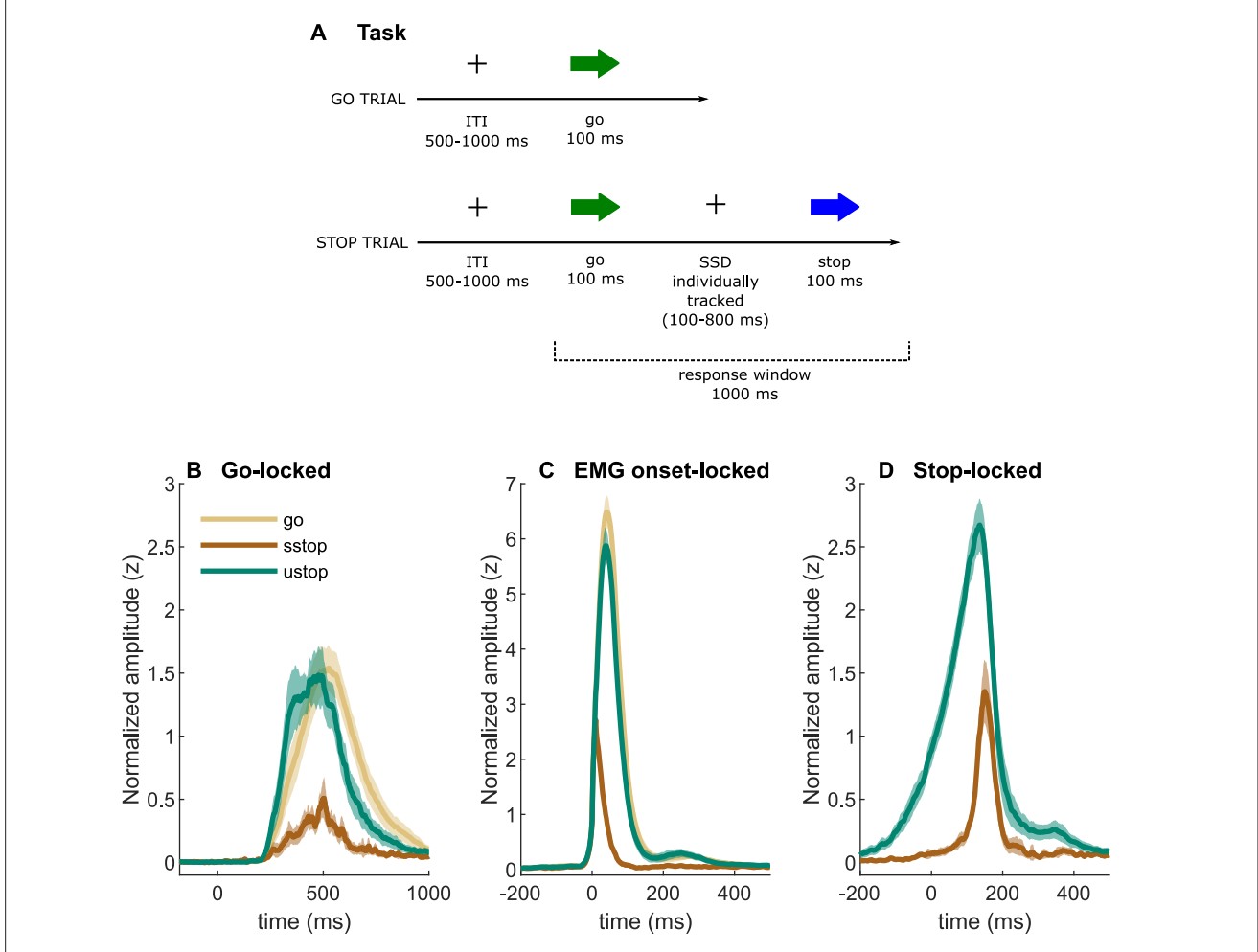

**Figure 1.** Task description and EMG time-courses. (**A**) Task depiction; (**B, C**) electromyography (EMG) time-courses, time-locked to (**A**) go signal onset, (**B**) EMG onset, and (**C**) stop signal onset. The lines depict averaged activity and the shaded areas depict 95% confidence intervals across participants. sstop – successful stop; ustop – unsuccessful stop.

EMG bursts were identified automatically using a threshold-based algorithm (see *Box 2* and Materials and methods for details). EMG bursts were detected in 99% of go trials and 98% of unsuccessful stop trials. Note that while 100% detection rate is expected in those trials, variations in signal quality (increased baseline muscle tone, unrelated movement artifacts, etc.) led to slight conservative bias in the detection algorithm. PrEMG was detected in 23% of successful stop trials.

Averaged EMG waveforms for each trial type are shown in *Figure 1* and the summary statistics are presented in *Table 2*. When time-locked to go signal onset, the trials with a registered button press showed large EMG activity (*Figure 1B*). Note that go-locked averaging over varied SSDs causes a smeared waveform for prEMG. We used Bayesian hypothesis testing to compare the EMG onset latencies and peak amplitudes between trial types. This confirmed that the EMG onsets were earlier in unsuccessful stop trials than in go trials (BF = $1.32 \times 10^{24}$) and that prEMG onset latencies were similar to the EMG onset latencies in go trials (BF = 0.28). When time-locked to the EMG onset (*Figure 1C*), the rising flanks of the EMG waveforms were remarkably similar across all trials but reached very different amplitudes. The peak amplitudes in successful stop trials were considerably smaller than in go (BF = $2.77 \times 10^{29}$) and unsuccessful stop trials (BF = $1.91 \times 10^{27}$). Interestingly, EMG amplitudes in unsuccessful stop trials were also smaller than those in go trials (BF = BF = $2.32 \times 10^{7}$), suggesting that the former were affected by stop signal processing despite the registered button press.

When time-locked to stop signal onset, a very clear prEMG waveform appeared (*Figure 1D*), which started declining at 165 ms after the stop signal. A similar decline also occurred in unsuccessful stop

**Table 1.** Behavioral and BEESTS parameters summary statistics.
RT = reaction time; sd =standard deviation; SSD = stop signal delay; SSRT = stop signal reaction time. *N* = 46. *One outlier was removed from the SSRT sd data due to an implausible value (378 ms).

| | Mean | sd | Median | min | max |
|---|---|---|---|---|---|
| **Behavior** | | | | | |
| Go accuracy (%) | 97.92 | 2.21 | 98.76 | 91.23 | 100.00 |
| Go errors (%) | 0.69 | 0.88 | 0.37 | 0.00 | 5.12 |
| Go omissions (%) | 1.40 | 1.95 | 0.73 | 0.00 | 8.04 |
| Go RT (ms) | 545.28 | 77.11 | 549.04 | 398.29 | 715.24 |
| SSD (ms) | 347.50 | 97.59 | 349.53 | 155.71 | 551.60 |
| SSRT (ms) | 198.99 | 29.63 | 202.00 | 120.05 | 263.97 |
| Stop accuracy (%) | 49.88 | 1.84 | 50.46 | 43.06 | 54.17 |
| Unsuccessful stop RT (ms) | 480.47 | 70.85 | 496.64 | 351.11 | 671.92 |
| **BEESTS parameters** | | | | | |
| Go failures (%) | 1.56 | 2.18 | 0.76 | 0.04 | 9.12 |
| Trigger failures (%) | 1.27 | 1.62 | 0.51 | 0.28 | 8.94 |
| Go RT (mean; ms) | 546.51 | 78.83 | 548.71 | 396.93 | 722.25 |
| Go RT (sd; ms) | 113.52 | 28.97 | 112.25 | 65.57 | 167.46 |
| SSRT (mean; ms) | 201.83 | 21.39 | 200.03 | 166.22 | 248.85 |
| SSRT (sd; ms)* | 30.70 | 13.38 | 27.69 | 14.94 | 86.55 |

The online version of this article includes the following source data for table 1:

**Source data 1.** Behavioral data.

**Source data 2.** BEESTS data.

trials, although they peaked slightly earlier at 99 ms after the stop signal (BF = $3.47 \times 10^{12}$). Speculatively, the steep decline of the falling flank seen in unsuccessful stop trials may suggest that unsuccessful stop trials were also affected by the processes triggered by the stop signal around the same time as the prEMG decline in successful stop trials.

We additionally quantified motor time, that is the difference between the EMG onset and the RT, and the rise time, that is the duration of the EMG burst from its onset to peak (relative to go signal). Even though the differences are numerically small (**Table 2**), the motor time was longer in unsuccessful stop trials than in go trials (BF = 5.94; replicating the finding of **van de Laar et al., 2014**). When go trials were divided into fast, medium, and slow trials (see paragraph about the assessment of contextual independence for details on this division), the motor times increased with increasing RTs, so that slow go-trial motor times were equivalent to unsuccessful stop trial motor times (BF = 0.16). The motor time results deviated somewhat in the replication dataset, which had generally much longer motor times at 118 ms. While motor times still increased with increasing RT in go trials, the unsuccessful stop trial motor times were equivalent with the medium part of the go distribution (BF = 0.39), resulting in equivalence also with the full distribution (BF = 0.17). The absolute rise times were also longer in the replication dataset (ranging between 30 and 60 ms for different conditions), but they were shorter in unsuccessful stop trials than in go trials in both datasets (discovery BF = $1.16 \times 10^6$; replication BF = $3.93 \times 10^5$). Altogether, these results indicate different response dynamics in unsuccessful stop trials and go trials, suggesting that unsuccessful stop trials may be affected by the stop process as well. However, this may be contingent on task procedure and even on the response device mechanics. Here, we used a response device with very little resistance to produce a button press, resulting in very small motor times and low proportion of trials with prEMG. Nonetheless, the replication dataset with longer motor times and a higher prEMG proportion still indicates different response dynamics in go and unsuccessful stop trials.

## Box 2. Detecting prEMG and extracting peak latency.

### EMG detection

The identification of EMG bursts is a signal detection problem. Given that the signal-to-noise ratio of EMG data is relatively high, simple threshold-based detection algorithms perform reasonably well. Basic preprocessing steps to further increase the signal-to-noise ratio may include high-pass filtering, rectification or a root mean square transformation, smoothing or low-pass filtering, baseline normalization, and the definition of a temporal window eligible for burst detection. We additionally normalize the full EMG time-courses to be able to keep the detection threshold constant across participants. Our processing pipeline is freely available at the Open Science framework. Alternatively, one may refer to the toolbox provided by *Jackson and Greenhouse, 2019*.

Simple threshold-based algorithms may, however, show declined performance in the presence of high baseline EMG activity or when crosstalk from different muscle groups is prevalent. In such cases, burst detection in the frequency domain may outperform detection in the time-domain (see *Atsma et al., 2018*; *Liu et al., 2015* for examples).

### Extracting peak latency

There are several ways to extract the prEMG peak latency. Here, we extracted the peak latency of an EMG burst from each trial in which prEMG was detected, and then averaged across trials to get a summary measure for each participant (*Box 2—Figure 1*, left upper panel). This method typically requires extra cleaning and restraints, discarding trials with elevated baseline muscle tone or irrelevant muscle artifacts outside of the expected response window. A more robust method is to first average the EMG activity time-point-wise across all successful stop trials with prEMG and then extract a single peak latency from the averaged waveform (*Box 2—Figure 1*, left middle panel). A third alternative is to extract the average peak latency from trials stemming from the most frequent SSD (the mode; *Box 2—Figure 1*, left lower panel). This has the closest resemblance to the SSRT estimation (as that is typically calculated at a single SSD, such as the mean SSD), but this method also has significantly fewer available trials, reducing the overall signal-to-noise ratio.

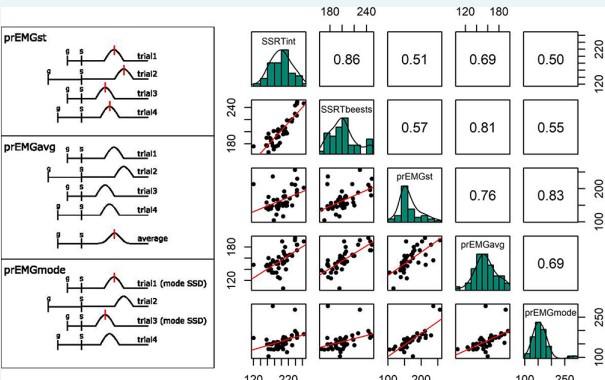

**Box 2—figure 1.** (Left panels) Schematics of partial response electromyography (prEMG) peak latency extraction methods on four trials, in which the 'bumps' represent rectified and smoothed EMG bursts, and the vertical red lines indicate the extraction points for latencies from the time-courses. In case of several red lines, the final estimate was averaged across the extracted latencies. G and s mark the onset of go and stop signal, respectively, with s also marking the zero time-point. (Right panels) Correlations between the prEMG peak latency and the SSRT, calculated by different methods. The diagonal shows the histograms for each variable together with the density functions, the lower diagonal shows the scatterplots, and the upper diagonal shows the Spearman correlation coefficients. SSRTint – integration method; SSRTbeest – BEESTS method; prEMG_st – extracted from single trials; prEMG_avg – extracted from the average waveform of all trials; prEMG_mode – estimated from the mode SSD.

We compared the three methods and found a good correspondence between them (Spearman *r* between 0.69 and 0.83). All three methods also correlated with the SSRT (*Box 2—Figure 1*, right panels). Peak latency estimates extracted from averaged waveforms appear more normally distributed than estimates from single trials, indicating that the former may represent the more robust estimation approach. However, as this method gives only a single summary measure per participant, it is not appropriate if single-trial analyses are of interest. The prEMG peak latency calculated using only the trials at mode SSD seems to be susceptible to outliers, likely because very few trials are used to derive the summary measure.

## Associations of prEMG peak latency and frequency with behavior

In many studies, the SSRT has been reported as a single behavioral outcome measure of the stop signal task, purportedly reflecting the speed of response inhibition. However, recent evidence indicates that both SSRTs and prEMG peak latencies are related to go RTs, potentially due to behavioral adaptations beyond response inhibition (*Huster et al., 2020*). We therefore investigated the correlational structure of the prEMG and other behavioral measures, both across individuals in the current study and across different experiments by performing a meta-analysis. Given that the major caveat of the prEMG is that it is only present in a portion of successful stop trials, we further investigated whether its peak latency is affected by the proportion of available trials.

Firstly, goRTs correlated strongly with the SSRTs ($r = -0.61$, $p = 1.62 \times 10^{-5}$) and moderately with the prEMG latencies ($r = -0.33$, $p = 0.027$). Secondly, goRTs correlated moderately with the prEMG detection frequency ($r = -0.35$, $p = 0.018$). The relationships between prEMG frequency and stopping latency indices were not significant (SSRT $r=0.23$, $p = 0.127$; prEMG peak latency $r = 0.27$, $p = 0.065$). In sum, persons who responded fast had more prEMG trials and longer stopping latencies, likely due to a strategic trade-off between going and stopping. Noticeably, this trade-off seems to be best captured by the associations between the goRT and the SSRT.

**Table 2.** Means (and standard deviations in brackets) of the electromyography (EMG) variables.

Motor time refers to the difference between the EMG onset latency and the reaction time. Rise time refers to the duration from EMG onset to its peak (both calculated relative to the go signal). AUC = area under the curve from onset to peak (relative to go signal). *Peak latencies in stop trials are time-locked to the stop signal onset. N = 46.

| Variable | Go | Succ. stop | Unsucc. stop |
|---|---|---|---|
| Count | 655 (25) | 23 (13) | 106 (5) |
| Percentage | 99 (2) | 23 (14) | 98 (2) |
| Onset latency (ms) | 487 (81) | 481 (91) | 422 (74) |
| Peak latency* (ms) | NA (NA) | 165 (31) | 99 (37) |
| Peak amplitude (z) | 12 (1) | 4 (1) | 11 (1) |
| Motor time (ms) | 57 (13) | NA (NA) | 58 (14) |
| Rise time (ms) | 52 (10) | 19 (5) | 48 (9) |
| AUC (z) | 142 (18) | 31 (12) | 125 (16) |

The online version of this article includes the following source data for table 2:

**Source data 1.** EMG summary data.

Next, we quantified these relationships across 21 different experiments or conditions as reported in 10 studies (*Figure 2*). Despite the large variability of goRTs across the studies (range 246 ms), the prEMG peak latency was surprisingly stable (range 35 ms) and averaged to 158 ms with an average detection frequency of 49% (see *Table 3* for more descriptive statistics). Given that most articles reported several experiments or task conditions, relationships between prEMG and behavior were tested using general mixed models with article as a random variable (but the relationships were the same if the average value across conditions and experiments was used for each study; *Figure 2E*). The strong negative association between go RTs and SSRTs seen across individuals was also evident across studies ($b = -2.08$, $t(19) = -2.65$, $p = 0.012$), as was the smaller association between the go RTs and prEMG peak latencies ($b = -1.33$, $t(11) = -2.21$, $p = 0.049$). Similarly, studies with faster go RTs reported higher prEMG detection frequencies ($b = -2.03$, $t(15) = -2.96$, $p = 0.010$). Interestingly, while every single study reported a positive correlation between the prEMG and the SSRT with an average correlation of $r = 0.64$ (standard deviation covering the range from 0.44 to 0.78 across studies after Student z- and retransformation; *Figure 2D*), this

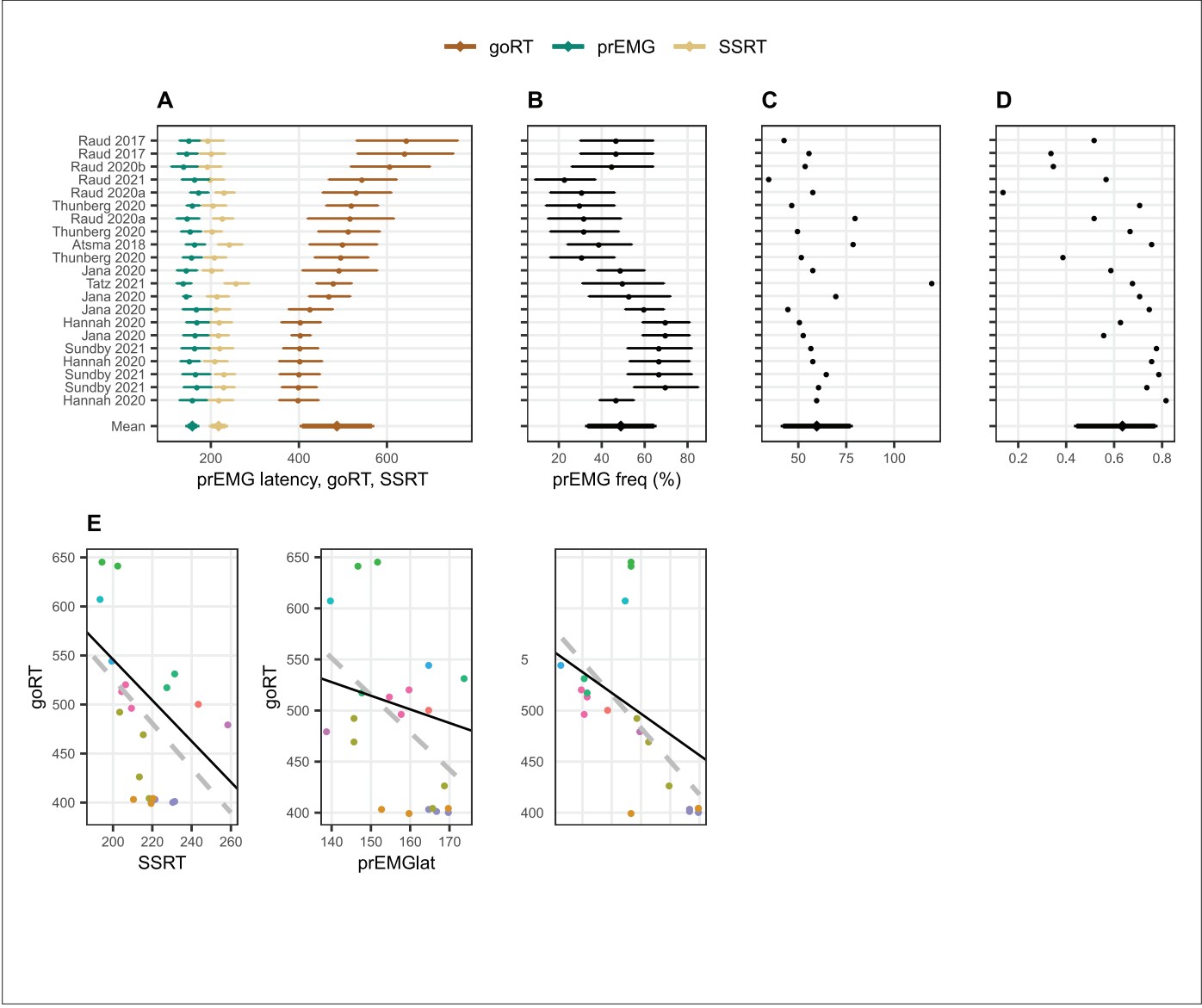

**Figure 2.** Meta-analysis of studies reporting partial response electromyography (prEMG) in stop signal tasks, which included 21 experiments or conditions reported in 10 independent studies. (**A–D**) Means and standard deviations (error bars) of each entry, ordered by go RTs. The error bars around the *y*-entry 'Mean' represent standard deviations across studies. (**E**) Associations between behavioral and EMG variables across studies. The dots reflect each separate condition/experiment study, color coded by the studies they originate from. The dashed gray lines represent bivariate regression lines, and the solid black lines represent the fixed effects of the general linear mixed model, accounting for the fact that single studies have more than one entry. The regression lines for the stop signal reaction time (SSRT)-prEMGlat plot are calculated excluding the outlier at top left. Note that the entry marked ***Raud et al., 2021*** represent the data from current study.

The online version of this article includes the following source data for figure 2:

**Source data 1.** Meta-analysis data.

relationship was not particularly strong across studies, reaching significance only after discarding an obvious outlier ($b = 0.34$, $t(11) = 2.29$, $p = 0.044$). Most importantly, while prEMG detection frequency varied considerably across studies (between 23% and 70% of stop trials), it was not associated with its peak latency ($b = 0.14$, $t(19) = 0.91$, $p = 0.375$).

## Comparison of prEMG peak latency and different SSRT estimates

There are several different ways to calculate the SSRT. The two state-of-the-art techniques are the integration method with replacement of go omissions (***Verbruggen et al., 2019***), and the Bayesian

**Table 3.** Meta-analysis behavioral and electromyography (EMG) values across 21 different conditions or experiments in 10 independent studies. prEMG = partial response electromyography; SSRT = stop signal reaction time.

|  | Mean | sd | Median | min | max |
| --- | --- | --- | --- | --- | --- |
| Go RT (ms) | 486 | 79 | 493 | 400 | 646 |
| SSRT (ms) | 217 | 16 | 216 | 194 | 259 |
| prEMG frequency (%) | 49 | 16 | 47 | 23 | 70 |
| prEMG peak/offset latency (ms) | 158 | 11 | 160 | 139 | 174 |
| SSRT and prEMG latency difference (ms) | 60 | 18 | 57 | 35 | 120 |

parametric approach (*Heathcote et al., 2019*; *Matzke et al., 2013*), which has been extended to incorporate go errors and failures to trigger both the go and the stop process (*Matzke et al., 2019*; *Matzke et al., 2017*). Where some studies show prevalent trigger failures and considerably faster BEESTS-SSRTs than integrations-SSRTs (*Skippen et al., 2020*; *Skippen et al., 2019*), others find a lower incidence of trigger failures and correspondingly little change in SSRT estimates (*Jana et al., 2020*). In line with the latter, we found the different SSRT estimates to be comparable in this dataset (BF = 0.34). Both were considerably slower than the prEMG peak latency (integration BF = $9.6 \times 10^5$, BEESTS BF = $2.6 \times 10^9$). Both SSRTs correlated moderately with the prEMG peak latency (integration $r$ = 0.51; BEESTS $r$ = 0.57 for prEMG derived from single trials, but see *Box 2* for differences in correlations with different prEMG peak latency estimations). Consistent with previous studies, the delay between the prEMG and the SSRT (integration) was 33 ms (sd = 33). This did not correlate with the prEMG detection frequency ($r$ = 0.04).

## Comparison of behavior- and EMG-based inhibition functions

Having established the basic summary characteristics and their (non-)correspondence with the SSRT, we set out to investigate how the prEMG relates to the framework of the horse race model of going and stopping. A feature of the stop signal task is that the SSD varies, most commonly due to an online tracking algorithm that adapts the SSDs to each participant's performance. The purpose of this is to capture the SSD where participants manage to stop their response for about 50% of the time. A key assumption of this approach is that the probability of unsuccessful stopping (p(button press|stop)) is low (approaching zero) at short SSDs, increases steeply at mid-SSDs, and approaches one at long SSDs. With EMG, this would correspond to the probability of producing EMG at any given stop trial (i.e., p(EMG|stop)), as this implies that the go process was at least initiated before stopping process. This probability thus includes all stop trials with detectable EMG, and is conceptually the sum of unsuccessful stop trials and successful stop trials with prEMG. Thus, by definition, the function of p(EMG|stop) should be shifted to the left from the function of p(button press|stop). In addition, assuming that the successful stop trials with prEMG are drawn uniformly from the distribution of successful stop trials, we would expect the two functions to run in parallel. However, preliminary evidence suggests that prEMG occurs more often at short SSDs

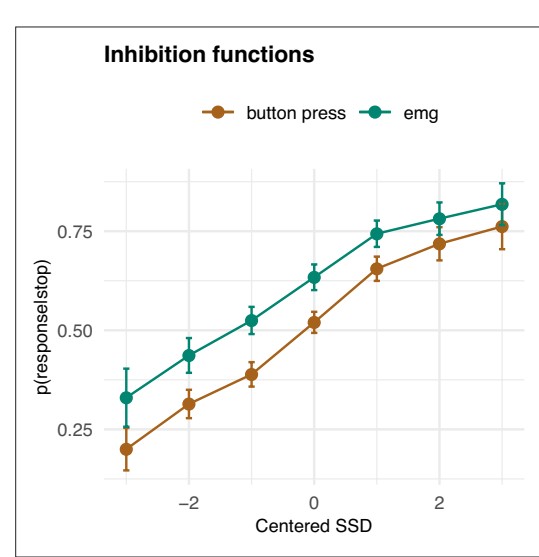

**Figure 3.** Inhibition functions. Error bars represent 95% confidence intervals. Note that the error bars are calculated across all trials and participants, ignoring that each participant has several data points.

The online version of this article includes the following source data for figure 3:

**Source data 1.** Inhibition functions data.

(*Coxon et al., 2006*; *de Jong et al., 1990*). Given that p(EMG|stop) represents the sum of unsuccessful stop trials and successful stop trials with prEMG, this would lead to larger differences between the two functions at short than in long SSDs.

The empirical inhibition functions are depicted on *Figure 3*. To capture the expected nonlinear shape of the functions, we fit a generalized additive mixed model (GAMM), predicting the probability of responding (p(button press|stop)) or producing EMG (p(EMG|stop)) despite a stop signal as a function of SSD (model $R^2$ = 0.64); this model thus also allows for the assessment of the interaction of response type (button, EMG) and SSD. The effect of SSD was significant (df = 3.18, $F$ = 213.91, p < 2 × $10^{-16}$), as was the parametric effect of response type ($b$ = 0.07, $t$ = 9.52, p < 2 × $10^{-16}$). Further, the interaction term was significant (df = 1.00, $F$ = 6.795, p = 0.009), as the difference between detecting a button press compared to an EMG response was larger at smaller SSDs. Since p(EMG|stop) contains p(button press|stop) as well successful stop trials with prEMG, these effects indicate that prEMG was more common in trials with short SSDs.

## Context independence of going and stopping

A key assumption of the horse race model is that the go and stop processes are independent of each other. In particular, the context independence assumption states that the stop process does not affect the primary go process, thus the distribution of the go-finishing times (estimated by go RTs) is identical in the go and stop trials (*Band et al., 2003*). We applied two tests of this assumption to both the behavioral and EMG data.

The most common test compares the mean go RT with the mean RT of unsuccessful stop trials. If response activation is identical in the go and stop trials, the unsuccessful stop trial distribution is censored at the right side; it thus follows that the mean RTs should be shorter in unsuccessful stop trials than in go trials. This was true in our data both for RTs (BF = 1.53 × $10^{24}$) and EMG onset latencies (BF = 1.32 × $10^{24}$).

However, a more stringent test directly compares the equivalence of the corresponding portions of the response time distributions in go and stop trials. Dividing the go-RT distribution at the percentile corresponding to p(button press|stop, SSD) gives an estimate of the distribution of trials for a given

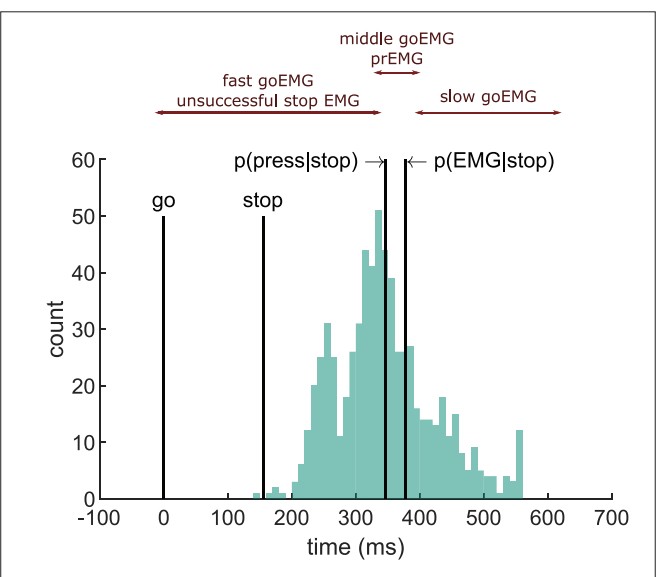

**Figure 4.** Depiction of the extension of the horse race model on electromyography (EMG) data extracted from an example participant.

The histogram depicts the distribution of EMG onsets in go trials, and the vertical lines mark (from left to right) the go signal onset, stop signal onset (mean stop signal delay [SSD]), percentile corresponding to the probability of producing a response and to the probability of producing EMG in stop trials. Assuming stochastic independence of go and stop trials, unsuccessful stop trials should correspond to the fast go trials (left side of the distribution between go signal and p(press|stop)), while partial response electromyography (prEMG) should correspond to the middle partition of the go distribution in between the p(press|stop) and p(EMG|stop).

**Table 4.** EMG onsets in go trials divided into fast (between go onset and p(press|stop)), medium (between p(press|stop) and p(EMG|stop)) and slow (> p(EMG|stop)) trials.
usEMG = unsuccessful stop EMG; prEMG = partial response EMG (successful stop trials).

|  | Mean | sd | Median | min | max |
|---|---|---|---|---|---|
| goEMG_fast | 401 | 67 | 410 | 260 | 555 |
| goEMG_medium | 506 | 80 | 500 | 347 | 678 |
| goEMG_slow | 608 | 108 | 582 | 416 | 844 |
| usEMG | 422 | 74 | 430 | 272 | 596 |
| prEMG | 481 | 91 | 474 | 290 | 701 |

The online version of this article includes the following source data for table 4:

**Source data 1.** prEMG onset data.

SSD in which the go process would or would not escape the stop process. Specifically, the portion at the left of this division are the fast go trials that would escape inhibition, and the portion on the right are slow go trials that would be stopped. We found that the mean unsuccessful stop trial RTs were not equivalent to the mean of the left side of the go-RT distribution (mean = 470 ms, sd = 72; BF = 1542). Since the means of the fast go-RT portions may misrepresent the central values, we repeated the analysis using median values instead. Here, we did find the participant-specific medians of the left side of the go-RT distribution equivalent to the unsuccessful stop trial RTs in the discovery set (BF = 0.53), but not in the replication dataset (BF = 34,849).

This test has been extended to the EMG onset latencies, deriving an additional division line using the percentile corresponding to p(EMG|stop, SSD) (*de Jong et al., 1990*; *Jennings et al., 1992*; *van Boxtel et al., 2001*). *Figure 4* explains the key expectations. Adhering to the independent horse race model, we expected EMG onsets in unsuccessful stop trials to match fast go-EMG onsets. Then, as the p(EMG|stop) is larger than p(button press|stop), there is a second division line derived from the percentile corresponding to p(EMG|stop), which is shifted to the right. The trials between those two lines should correspond to successful stop trials with prEMG. Consequently, the mean go-EMG onset latencies from trials between these two division lines should be equal to the prEMG onsets. Lastly, the go-EMG onsets to the right of the second dividing line should be slower than the prEMG onsets.

The EMG onset latencies, averaged over all trials within each partition of the go-EMG distribution, are listed in (*Table 4*). The data indicate a deviation from the independence assumption already for the left side of the distribution, as the average EMG onset in unsuccessful stop trials was slower than the average EMG onset in fast go trials (BF = $2.67 \times 10^9$). Secondly, the average prEMG onsets were faster than the corresponding average onset of the middle partition of the go-EMG distribution (BF = $2.17 \times 10^4$), which contrasts the prediction that these should be roughly equivalent. Lastly, the slow go-EMG onsets occurred later than the prEMG onsets, as expected (BF = $2.81 \times 10^{21}$). These patterns remained the same if medians were used instead of means. To confirm that the results are not affected by sampling differences due to having more go trials than stop trials, we repeated this analysis by randomly sampling matching numbers of fast go and medium go trials for each participant and doing *t*-tests on these subsamples (on average 105 and 23 trials, correspondingly) 1000 times. The pattern of results stayed the same every single time. These results indicate that extending the horse race model to the EMG is not applicable and, consequently, estimating the SSRT based on go-EMG distributions should be avoided.

Even though the EMG onset patterns do not seem to be in accordance with the assumption of the horse race model, it is important to determine the prEMG characteristics across all SSDs to establish whether stopping estimates based on the prEMG are representative of the overall stopping behavior. Based on previous findings, we expected the prEMG frequency to decline and its amplitude to increase with increasing SSD. Further, as go RTs tend to correlate positively with the SSDs, we expected that the prEMG onset latencies (calculated relative to the go signal) would increase with SSDs as well. Note that the prEMG peak latency, in contrast, is calculated relative to the stop signal and thus reflects the speed of the stop process. Critically, it has never been asked whether the prEMG peak latency remains stable at different SSDs.

We fit four GAMMs to predict the prEMG probability, onsets, amplitudes, and peak latencies from the SSDs (*Figure 5*). As expected, prEMG probability decreased with the SSD (df = 1.92, *F* = 24.28, p = $2.47 \times 10^{-10}$, $R^2$ = 0.09). Similarly, we found increased EMG onsets with increasing SSD (df = 2.60, *F* = 420.7, p < $2.00 \times 10^{-16}$, $R^2$ = 0.36). The prEMG peak latencies decreased with SSD (df = 2.29, *F*

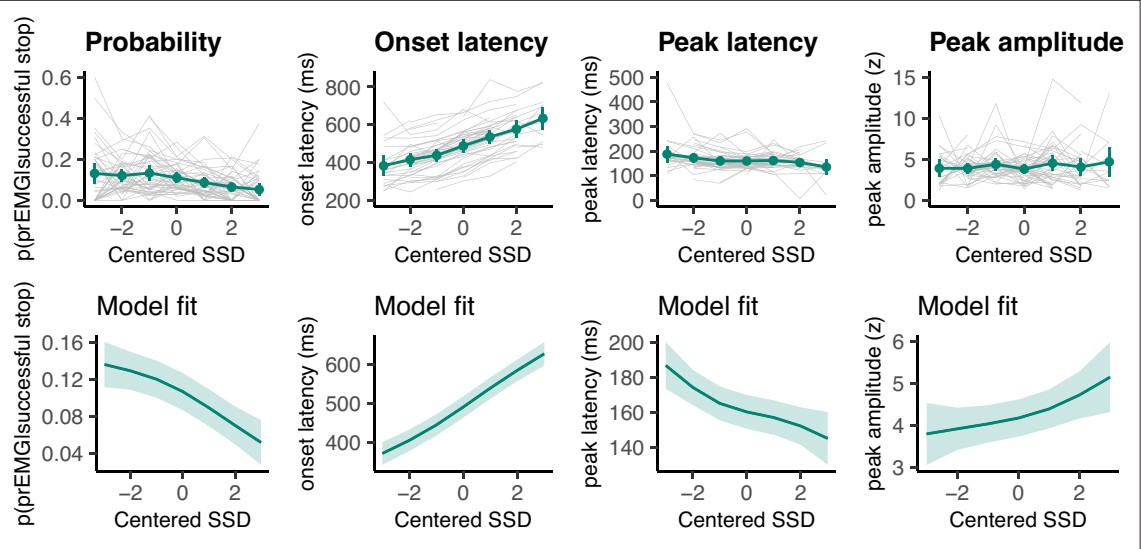

**Figure 5.** Partial response electromyography (prEMG) probability, onset latency, peak latency, and peak amplitude. The upper panels represent empirical data across all participants. Green lines represent the group average with 95% confidence intervals and gray lines represent each participant. The lower panels depict the model fit of the generalized additive models with 95% confidence intervals. Note the different *y*-axes between empirical data and model fits.

The online version of this article includes the following source data for figure 5:

**Source data 1.** SSD-centered prEMG frquency data.

**Source data 2.** SSD-centered prEMG onset latency data.

**Source data 3.** SSD-centered prEMG peak latency data.

**Source data 4.** SSD-centered prEMG peak amplitude data.

= 11.77, p = 9.71 × 10⁻⁶, $R^2$ = 0.06), indicating that stopping may be slower at short SSDs. PrEMG amplitudes increased slightly with longer SSDs, with the smooth term at the border of significance (df = 1.66, *F* = 4.16, p = 0.056, $R^2$ = 0.02).

## prEMG peak latency as a function of time on task

Another important question is whether prEMG remains stable across the experiment. It is possible that inhibition becomes automatic over time and loses its cognitive control element (***Best et al., 2016***; ***Verbruggen et al., 2014***; ***Verbruggen and Logan, 2008***). Consequently, prEMG peak latency may decrease after practice. We tested this by fitting a GAMM model predicting prEMG peak latency by trial order while controlling for SSD (model $R^2$ = 0.04), but we found no evidence that the trial order would affect prEMG peak latency (df = 1.173, *F* = 0.11, p = 0.843). We also tested whether the trial order affected prEMG frequency by predicting the existence of prEMG in a given trial from the trial order using logistic regression (while controlling for the SSD), but this model yielded no trial order effects either (df = 1.00, $\chi^2$ = 0.87, p = 0.350). Note that these results do not exclude that stopping may be automatic to begin with nor that automatism affects underlying processing (***Raud et al., 2020b***; ***Verbruggen et al., 2014***), but simply indicates that stopping performance did not change with practice in a single session.

## Single-trial associations of prEMG onset latency, peak latency, and area under the curve

A clear advantage of the prEMG is that it allows to study the dynamics of going and stopping at a single-trial level. A simplified set of predictions outlining different dependencies between going and stopping are depicted in ***Figure 6***, based on prEMG onset latencies (go-locked), peak latencies (stop-locked), and areas under the curves (AUC) from onset to peak. The AUC integrates the latency and amplitude of each prEMG burst and was calculated as the cumulative sum of point-wise amplitudes from onset to peak latency. Note that these scenarios are constructed assuming stable SSDs. This

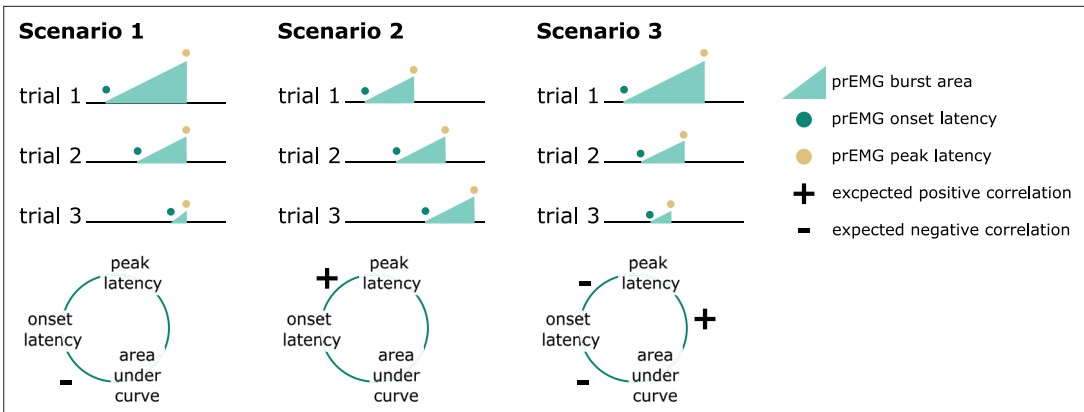

**Figure 6.** Predicted directions of single-trial correlations between electromyography (EMG) onset latency, peak latency, and area under the curve (calculated as the cumulative sum of point-wise amplitudes from onset to peak latency) under three different scenarios.

simplification helps to delineate the expected outcomes for different a priori hypotheses regarding the dependencies between going and stopping at the single-trial level.

Scenario 1 represents the independent horse race model, where there is considerable variability of the go process, but a constant and independent stop process. In this case, earlier prEMG onset leads to a larger AUC, but peak latencies are stable and independent of the onsets and AUCs.

Scenario 2 represents a case where delayed prEMG onset is associated with delayed prEMG peak latencies (as reported in *Raud et al., 2020a*), resulting in stable AUCs. This could be due to instances like competing resources for go and stop processes or attentional lapses that delay overall processing.

Scenario 3 represents a different dependency between the go and stop processes, in which more potent motor responses (i.e., earlier onsets and larger AUC) are associated with delayed stopping. This would result in a negative relationship between prEMG onset and peak latencies and between the onset latencies and AUCs, but a positive relationship between the peak latencies and the AUCs.

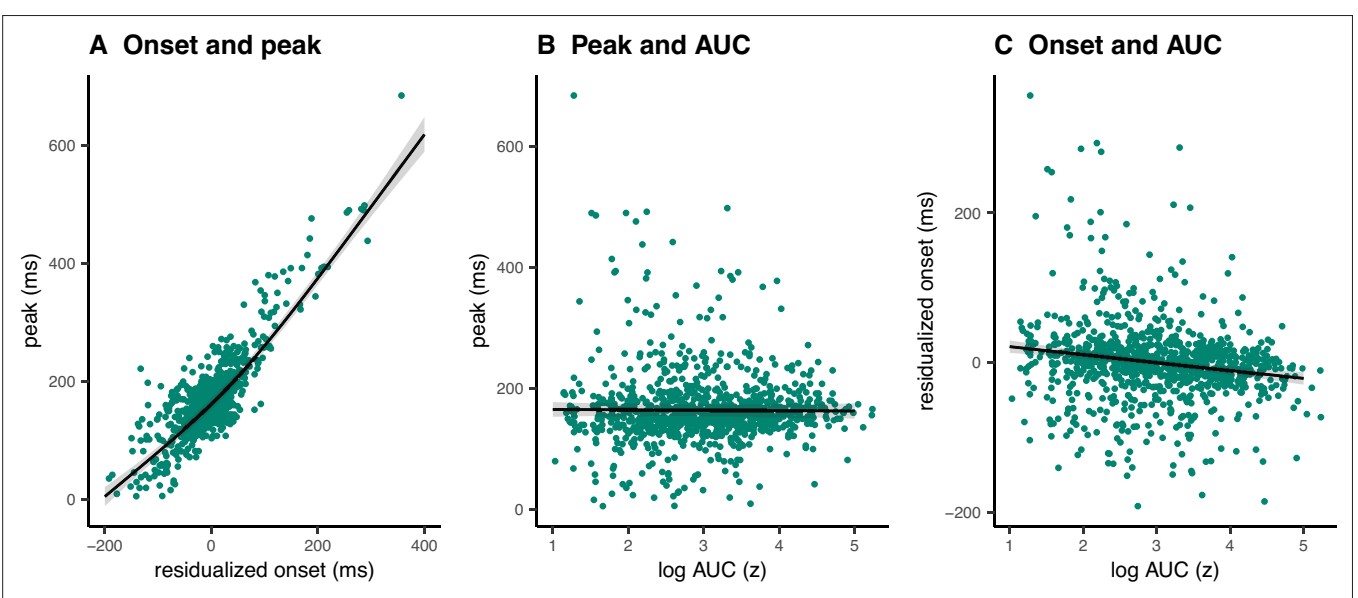

**Figure 7.** Single-trial associations between partial response electromyography (prEMG) onset latency, peak latency, and area under curve (AUC). The dots represent single trials from all participants. The black lines represent the fixed effect model fits from generalized additive mixed models, where the individual variability was modeled as separate intercepts for each participant. The shaded areas represent 95% confidence intervals.

The online version of this article includes the following source data for figure 7:

**Source data 1.** prEMG single trial data.

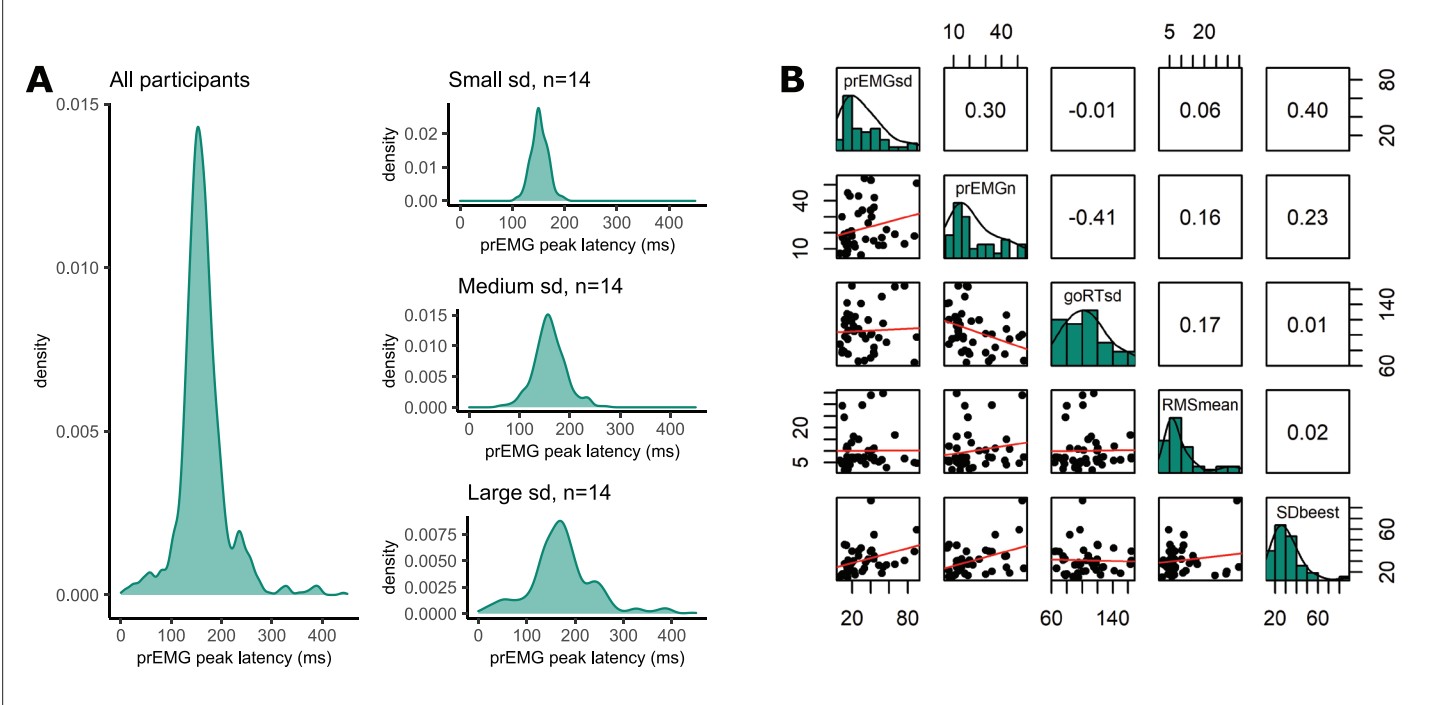

**Figure 8.** Variability of partial response electromyography (prEMG) peak latency. (A) Density functions of prEMG peak latency across all trials and participants (left panel) and separately for group of participants with low, medium, and large individual standard deviations (sd; right panels). (B) Correlation matrix between individual prEMG standard deviations (prEMGsd), number of prEMG trials (prEMGn), go reaction time standard deviations (goRTsd), average root mean square EMG amplitudes during the intertrial interval −200 to 0 before go stimulus onset (RMSmean), and standard deviations of the stop signal reaction time (SSRT) calculated through the BEESTS method (SDbeest). The diagonal shows the histograms for each variable together with the density functions, the lower diagonal shows the scatterplots, and the upper diagonal shows the Spearman correlation coefficients. prEMG reliability.

The online version of this article includes the following source data for figure 8:

**Source data 1.** prEMG variability data.

We tested these scenarios by fitting GAMMs predicting (1) peak latency from onset latency, (2) peak latency from AUC, and (3) onset latency from AUC. Onset latencies and SSDs correlated highly at the single-trial level ($r = 0.88$). To account for this, we predicted onsets from the SSDs for each participant and used the residuals instead of original onset values. AUCs were log transformed since the original values were extremely skewed toward low values.

The data showed evidence for scenario 2 (*Figure 7*) with a strong positive relationship between onset and peak latencies (estimated degrees of freedom [edf] = 3.47, $F = 1957$, $p < 2 \times 10^{-16}$, $R^2 = 0.72$). There was also a small negative association between onset latencies and the AUCs (edf = 1.00, $F = 29.25$, $p = 7.93 \times 10^{-8}$, $R^2 = 0.03$). In the discovery dataset, there was no relationship between peak latencies and AUCs (edf = 1.00, $F = 0.09$, $p = 0.77$, $R^2 < 0.001$), while the replication dataset showed a very small negative association (edf = 1.00, $F = 12.96$, $p = 0.0003$, $R^2 < 0.001$). As such, there was strong evidence for dependencies between going and stopping at the single-trial level, where delayed going was associated with delayed stopping. Note that none of these relationships were significant at the group level (i.e., taking the mean for each variable for each participant and fitting the same models), thus these results apply for single-trial data only.

## prEMG variability

Until now, most previous studies focused on a single estimate of the peak latency of the prEMG. However, prEMG allows for the calculation of the standard deviation of the peak latency for each individual; we can therefore empirically capture the variability of single-trial stopping latencies for each individual (*Jana et al., 2020*). A caveat here is that that we detect prEMG only in a fraction of the trials, hence the variability estimates are truncated and likely biased. Nonetheless, the prEMG

**Table 5.** Split half reliability estimates for prEMG characteristics, including mean, standard deviation, and percentiles derived from the permutation distributions that correspond to the 95% confidence intervals.
RMS = root mean square; AUC = area under the curve from onset to peak.

|  | Mmean | sd | 2.5th | 97.5th |
|---|---|---|---|---|
| Onset latency (ms) | 0.94 | 0.01 | 0.91 | 0.97 |
| Peak latency (ms) | 0.87 | 0.05 | 0.75 | 0.94 |
| Peak RMS amplitude (z) | 0.51 | 0.13 | 0.22 | 0.71 |
| AUC (z) | 0.48 | 0.11 | 0.24 | 0.68 |

The online version of this article includes the following source data for table 5:

**Source data 1.** Reliability data.

standard deviations show a considerable variability across participants and may therefore be informative of individual differences (*Figure 8A*). The average individual standard deviation was 31 ms (group-level sd = 21) and varied from 7 to 90 ms. Note that four outliers with values above two interquartile ranges were discarded from this analysis (see *Appendix 2—figure 1* for visualization of the outliers).

To understand the variability measure better, we first considered whether the between-participant variability could be driven by nuisance factors, such as the number of prEMG trials per person or the amount of baseline noise in the EMG. Secondly, it is possible that the variability of the stopping latency simply reflects variability in general processing, which can simply be assessed through the analysis of plain go responses without any need for a stopping variability measure. Therefore, we correlated the variability of the prEMG peak latency with the variability of the go RTs, where a low correlation would indicate that stopping variability gives information beyond the variability seen in go responses. The correlation matrix is depicted in *Figure 8B*. Note that we are reporting all correlation coefficients from this exploratory analysis with p values uncorrected for multiple comparisons, as the effect sizes are more informative about potential relationships than their significance.

The individual variability in prEMG peak latency correlated moderately with the number of prEMG trials ($r = 0.30$, $p = 0.05$), but showed no associations with baseline EMG noise ($r = 0.06$, $p = 0.72$) or go-RT variability ($r = -0.01$, $p = 0.94$). Lastly, we replicated the moderate positive correlation reported by *Jana et al., 2020* between the prEMG variability and the variability of the SSRT calculated using the BEESTS method ($r = 0.40$, $p = 0.01$). The correlational structure did not change if four outliers were included in the data, nor when the analysis was confined to participants with more than 10 prEMG trials. The data thus suggest that an individual's standard deviation of the prEMG peak latency provides complementary information on individual differences in stopping behavior.

## prEMG reliability

An important question for individual differences research is whether we can reliably estimate the prEMG characteristics despite the varying number of available trials per participant. We therefore estimated the split-half reliability of the prEMG-based latency and amplitude measures (*Table 5*), using a permutation-based split-half reliability approach (*Parsons et al., 2019*). Using all available trials and participants, we found high reliability for latency measures (0.94 and 0.87 for the prEMG onset and peak latency, respectively) but somewhat lower reliability for the prEMG peak amplitude and AUC (0.51 and 0.48, respectively). The same pattern was observed in the replication dataset, with even higher reliability coefficients for latency measures (0.98 and 0.92 for onset and peak latency, respectively).

A critical question for further research is how many trials and participants are necessary for reliable prEMG estimates. For instance, it may seem reasonable to exclude participants with few prEMG trials from further analysis. Yet, this could be harmful for the overall power due to increased standard errors across participants. We therefore repeated the split-half reliability analysis, sampling the lowest number of trials available for all participants and increasing the number of trials while excluding participants if they did not have the necessary number of trials available. This procedure resulted in incremental reliability estimates starting with 46 participants with 6 trials and ending with 12 participants with 32 trials (*Figure 9*). Interestingly, the reliability estimates for prEMG onset latency remained high (above 0.8) no matter the reduction in the number of trials and/or participants. prEMG peak latency reliability increased with the number of trials and reached a plateau around 0.77 after around 13 trials (with increased confidence intervals as the number

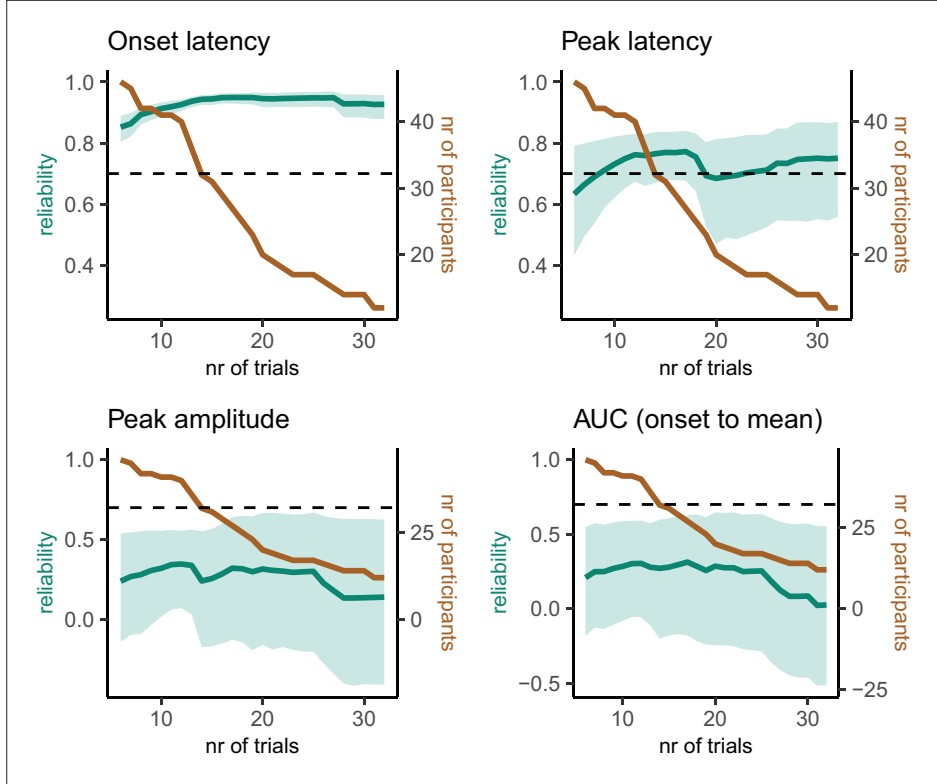

**Figure 9.** Reliability estimates as a function of number of trials (green) and number of participants (brown) for partial response electromyography (prEMG) latency- and amplitude-based measures. Green shaded areas represent 95% confidence intervals for the reliability estimates, and the black dashed horizontal line represent a reliability value of 0.7 for reference. Note the different *y*-axes for latency and amplitude measures. AUC = area under the curve from prEMG onset to peak.

The online version of this article includes the following source data for figure 9:

**Source data 1.** prEMG reliability data.

of participants decreased). As before, the reliability estimates for amplitude measures did not show good reliability, with values hovering around 0.25 and decreasing even further with fewer participants.

In sum, we conclude that the latency measures of prEMG have overall good reliability, while amplitude-based measures do not. While reliability measures are affected by the specifics of each dataset, our data suggest an optimal minimal number of trials to be around 15, and more is always better. However, the reliability estimates were highest when all participants (even those with less than 10 trials) were included. Thus, excluding participants based on low number of available trials is not strictly necessary and may cause more harm by increasing the group-level standard errors and thereby reducing statistical power.

## Discussion

We analyzed prEMG from a standard stop signal task to investigate its utility and reliability as a marker of action stopping. We compared it thoroughly to the best-established marker of stopping latency, the SSRT. Further, we evaluated whether prEMG supports the horse race model that underlies the SSRT estimation in nearly all studies using the stop signal task. We also provided practical tips for EMG data acquisition and analysis (*Box 2* and *Box 3*). We found that prEMG provides unique information about stopping and that assumptions of the horse race model may not regularly be met, questioning the broad and often undifferentiated use of the SSRT as a single indicator of stopping performance.

## Box 3. Practical tips for data acquisition.

**Recording systems**

EMG can be recorded by dedicated systems, but also by using EEG equipment that many human neurophysiology labs already own. In fact, if simultaneous EEG–EMG measures are of interest, it is preferable to record them by the same system so that both signals are synchronized on acquisition. Many EEG systems include a limited number of dedicated channel pairs that allow for the recording of activity referenced against each other instead of a referencing against the common EEG reference. Those channels can be used to record from electrodes placed on the skin above muscles to capture EMG. Alternatively, EMG electrodes can be referenced against the common EEG reference during acquisition and then re-referenced offline to get a single bipolar EMG channel. Note though that a major caveat of this procedure is that a larger distance between the active EMG electrodes and the EEG reference increases the chance of amplifier saturation, rendering the EMG signal useless.

**Muscle choice and movement direction**

The signal-to-noise ratio of EMG bursts is highly dependent on the combination of target muscle, electrode placement, and movement direction. Most studies have recorded from thumb muscles (abductor *pollicis brevis*), index fingers (*first dorsal interosseous*), or a combination of both, although head, neck, and biceps muscles have been used as well (***Atsma et al., 2018***; ***Goonetilleke et al., 2010***). The most intuitive movements (for pushing a button, for example) do not necessarily produce the largest signal-to-noise ratio in EMG recordings. Therefore, we recommend testing various ways of producing a movement before starting the experiment. For APB, the strongest signal is produced by moving the thumb down while keeping the rest of the fingers flat on a surface (it may thus be necessary to elevate the rest of the hand on a book or a block so that downward thumb movements become possible). For the index finger FDI, moving the finger horizontally while keeping the palms flat on the table gives the best results, sometimes outperforming the APB measurements (***Coxon et al., 2006***). Even though such movements may be unusual for the participants at first, they typically learn fast after some training. Gamified instructions may help to speed up the learning process (e.g., ***Quoilin et al., 2016***).

**Avoiding spurious muscle tension**

Even though EMG bursts related to movement are large and clear, detection algorithms may struggle with exact onset and peak latencies if a lot of baseline muscle tension is present. This can be reduced significantly by letting the participants rest their arms comfortably on a pillow, and by reminding them before and during the task to relax their hands. Such verbal reminders typically help even if the participant is not aware of any muscle tension. One potential problem is that specific instructions to relax the hands may interfere with task-related tonic EMG fluctuations, but to the best of our knowledge, such effects have not been investigated systematically.

## The independent horse race model and prEMG

Several results from our analysis indicate incompatibility of the prEMG with the horse race model. On the one hand, we replicated the results from *de Jong et al., 1990*, who demonstrated that prEMG trials correspond to trials in the go-trial distribution that are fast enough to initiate muscle activity, but slow enough to not outrun the stopping process. On the other hand, neither unsuccessful stop trials nor successful stop trials with prEMG matched the corresponding partitions of the go-EMG distributions. This indicates that the horse race model framework should not be extended to the go-EMG distribution. Furthermore, prEMG peak latency varied with the SSD, indicating that stopping latency is not stable. Lastly, the single-trial analysis indicated a major violation of the horse race model independence assumption, as the go process (measured by the go-locked prEMG onset latency) and the stop process (measured by the stop-locked prEMG peak latency) share about 70% of variance if the SSD is accounted for. Together, these results add to the increasing pool of evidence which suggests that

the independent horse race model is not an appropriate approximation of the stopping performance in practice.

## Comparison of the prEMG peak latency and the SSRT

There are three main findings when comparing the prEMG with the SSRT. First, the prEMG peak latency occurs earlier than the SSRT. Second, the prEMG correlates moderately to highly with the SSRT. Third, prEMG frequency and latency distributions show that the horse race model assumptions do not hold in practice, particularly in trials with short SSDs. This begs the question of whether prEMG and SSRT reflect the same underlying mechanism, and if not, how should these measures be interpreted?

The most parsimonious explanation would be that the prEMG and the SSRT capture the same conceptual construct – action stopping – but they simply reflect different stages in the cascade of events. The prEMG peak latency leads the SSRT by 60ms with an average correlation of $r = 0.64$ across studies. The 60ms delay between the myographic (prEMG peak latency) and behavioral (SSRT) measures may reflect the final ballistic component of the movement, the so-called point of no return, after which initiated movements are unaffected by any higher-order control mechanism (*Osman et al., 1986*). In this case, the prEMG may simply reflect the inertia in the neuromuscular system and response device presiding after the prepared movement has been stopped in the cortex (*Gopal and Murthy, 2016*; *Jana et al., 2020*). However, several lines of evidence challenge this explanation. First, such 'leakage' EMG has been shown to have a milder rising slopes compared to the normal go-EMG responses (*McGarry and Franks, 2003*; *McGarry and Franks, 1997*). In the current study, the EMG onset slopes were identical for all trial types (*Figure 1B*). Second, even with a moderate-to-strong association across all studies, correlation of prEMG and SSRTs varies considerably between studies. With explained variances ranging from about 10% to 60%, there remains plenty of unexplained variance even for the studies with the highest correlations. Third, the correlational structure with other behavioral and electrophysiological measures derived from the stop signal task differ greatly between the prEMG and the SSRT, where the latter exhibits a correlational structure indicative of an association with a trade-off between fast going and accurate stopping (*Huster et al., 2020*).

To account for the observed differences between the prEMG and the SSRT, we propose that both reflect the latency of action stopping to some degree, but that the SSRT is additionally affected by behavioral strategies and is thus more dependent on the task context. In support of this, the SSRT correlates negatively with go RTs, both across individuals in a single study, as well as across different studies. Whereas the prEMG detection frequency is also affected by task characteristics, evidenced by the correlations with go RTs, the primary outcome of interest, the peak latency, is relatively stable across different studies.

In sum, the prEMG peak latency captures action stopping speed and its variability across single trials. The SSRT likely captures action stopping to some degree but it is further affected by strategic behavioral adaptations and inaccuracies due to violations of the horse race model's assumptions.

## Generalizability of the prEMG peak latency

The generalizability of the prEMG is potentially hampered by the fact that it appears only in a fraction of trials. The prEMG latency distribution is a truncated representation of the distribution of the underlying stopping process, lacking trials in which actions were stopped cortically before muscle activation or in which go process was not initiated in the first place. The prEMG distribution is thus affected by two interacting processes – action generation and action stopping. The current dataset does not allow us to test which of these processes is dominating.

To estimate the potential effect of the missing trials, we tested whether the prEMG frequency is associated with other behavioral and EMG measures across participants and studies. First, participants and studies with faster RTs produced more prEMG trials, but there was no relationship between the prEMG frequency and peak latency. Of note, faster RTs were associated with longer stopping latencies, and this relationship was particularly strong for the SSRT. Second, whereas prEMG was detected only in a minority of 20% trials in our data, most of our results were replicated in an independent dataset where 60% of successful stops contained prEMG. Third, the split-half reliabilities of the prEMG latency measures were high even for a small number of trials. Altogether, the fact that the

prEMG distribution is truncated does not invalidate its use as a marker of action stopping, since the peak latencies are relatively stable despite the varying number of trials available for its estimation.

Nonetheless, as the contribution of the missing prEMG trials is unknown, it may still be argued that the available trials do not represent 'typical' stop trials. However, stopping actions may be achieved by different underlying mechanisms across different tasks, as well as within the same task across different trials. As such, whether there exists a 'typical stop-trial' is an empirical question that the prEMG can provide answers to. This can be achieved by (1) optimizing task design to produce more prEMG trials, for example by encouraging fast responding, using a response device with stiff buttons, or using an anticipated response task that discourages response slowing (*Coxon et al., 2006*; *MacDonald et al., 2014*); (2) requiring multicomponent movements and/or using antagonist muscle onsets as proxy of stopping latency in each trial (*Goonetilleke et al., 2012*; *Goonetilleke et al., 2010*; *De Havas et al., 2020*); (3) combining EMG with other parametric approaches in which action stopping can be measured on a continuous scale, for example measuring movement trajectories (*Atsma et al., 2018*; *De Havas et al., 2020*) or response force (*Nguyen et al., 2020*). If properly validated, parametric approaches may eventually outperform EMG measures, especially in clinical samples where obtaining clean EMG signals may be challenging.

Another unsettled question regarding the generalizability is whether prEMG trials could be qualitatively different from other successful stop trials. *McGarry and Franks, 1997* designed a stop signal task with untypically short SSDs, and found that they could differentiate between trials with early and late stopping mechanisms based on the shape of the prEMG rising slope (see *Box 1* about interrupted response for details), although it has been disputed that the same pattern could result from the differential recruitment times of motor neurons with different sizes (*van Boxtel and Band, 2000*). Further, EEG studies have found that prEMG trials show decreased lateralized readiness potentials and increased frontal negativities (*de Jong et al., 1990*; *van Boxtel et al., 2001*). Increased frontal negativities have also been related to awareness of the prEMG in response conflict tasks (*Ficarella et al., 2019*). However, whether these cortical changes reflect qualitatively different mechanisms or merely a continuous variation in the strength of the underlying processes is unresolved (*van Boxtel et al., 2001*).

Finally, does prEMG latency generalize across different contexts? While it has been shown that the SSRT becomes increasingly unreliable in more complex stop signal tasks and its use is thus discouraged (*Bissett and Logan, 2014*), prEMG can still be recorded in such task settings (*Raud et al., 2020a*; *Raud and Huster, 2017*). Further, (pr)EMG is not limited to the stop signal task, but can be recorded in any task that requires overt responses, as well as tasks that require more complex movements activating several muscle groups (*Tao et al., 2018*). As such, EMG can provide highly sought-after physiological measures that are comparable across different contexts and is currently the most useful approach for investigating putative response inhibition in tasks that rely on the absence of overt behavior as the primary outcome measure. For example, one study using this approach found that the prEMG peaked significantly later in the go/no-go task than in the stop signal task. This was also paralleled by differences in cortical activity, suggesting that these tasks may recruit different mechanisms (*Raud et al., 2020b*). Similarly, *Tatz et al., 2021* showed that EMG activity declines after any infrequent salient stimulus, highlighting the interdependence between attentional capture and inhibitory control constructs. Thus, comparisons of prEMG characteristics and their driving mechanisms in different contexts provide a rich ground for future research on motor and cognitive control.

## Response inhibition and prEMG

While prEMG can be considered an assumption-free physiological measure in its implementation, it is not free from assumptions in its interpretation. The most urgent unresolved question is whether prEMG captures the latency of inhibitory mechanisms in the brain. On the one hand, prEMG decline unequivocally indicates stopping of muscle activity, so it is safe to presume that the peak latency represents the latency of action stopping. On the other hand, the actual mechanisms that cause this EMG decline cannot be derived from prEMG alone. Alternative explanations to cortical active inhibition mechanisms include activation of antagonist muscles triggering peripheral lateral inhibition, an initial subthreshold activation of the go response, or cessation of the invigoration of the go response by the motor system. Several studies have investigated whether stopping of specific muscle activity could be caused by the activation of antagonist muscles and found that agonist stopping occurs

before antagonist activation; thus, the former cannot be caused by the latter (*Atsma et al., 2018*; *Corneil et al., 2013*; *Goonetilleke et al., 2012*; *Goonetilleke et al., 2010*; *Scangos and Stuphorn, 2010*). It is also unlikely that prEMG represents initial subthreshold go activations, as such stimulus-locked responses are typically <100 ms after stimulus onset (*Pruszynski et al., 2010*), while prEMG onsets are closer to 300–500 ms. However, it remains to be seen whether prEMG declines because of a reduction in response maintenance (e.g., by restoring the inhibitory state of the cortical-basal ganglia–thalamic loop during rest), or because of an active inhibition mechanism that blocks ongoing motor activity (e.g., by blocking the direct pathway output by increased inhibition via the subthalamic nucleus, or by direct inhibitory influences on M1; e.g., *Hynd et al., 2021*). Complementary simultaneous data from brain imaging and neural stimulation studies are necessary to understand the neural mechanisms that cause the halt in muscle activity.

## Conclusions

Based on the synthesis of previous work together with a comprehensive analysis of prEMG characteristics in an independent dataset, we conclude that prEMG is a useful and reliable measure of action stopping. PrEMG is not limited to specific muscles, stimulus modalities, or task contexts, so it provides a comparative physiological measure for a range of different scenarios. Even though there are some similarities between the SSRT and the prEMG, these two measures should not be used interchangeably, but rather complementarily. The SSRT benefits from decades of research and is therefore well characterized across different contexts and sample populations. Yet, it is founded on assumptions that are not always met in empirical data, and it seems to be more strongly affected by behavioral strategies than the prEMG. In contrast, prEMG is directly observable and provides action stopping latencies at a single-trial level. Its peak latency is relatively stable even though design choices may limit the number of available trials. Importantly, neither the SSRT nor the prEMG peak latency is an unambiguous marker of response inhibition in the brain, but since the prEMG is directly observable, it has stronger potential for guiding research on brain mechanisms underlying action stopping.

# Materials and methods

**Key resources table**

| Reagent type (species) or resource | Designation | Source or reference | Identifiers | Additional information |
|---|---|---|---|---|
| Software, algorithm | E-prime | Psychology Software Tools, Pittsburgh, PA | Version 2.0 | |
| Software, algorithm | Matlab | The Math Works, Inc *MATLAB* | 9.4.0.949201 (R2018a) | |
| Software, algorithm | Eeglab | *Delorme and Makeig, 2004* | 14.1.1b | |
| Software, algorithm | R | *R Core Team, 2019*; | 3.6.2 (2019-12-12) | |
| Software, algorithm | BEESTS; Dynamic Models of Choice | *Matzke et al., 2019*; *Matzke et al., 2013* *Heathcote et al., 2019* | BEESTS2-GF | |

## Sample

Data were analyzed from 46 healthy young adults (mean age = 23.7, sd = 5.5, 30 females, 3 left-handed). Initially, data were collected from 52 participants, but four were discarded due to technical issues with the response device and another two were discarded as statistical outliers (above two interquartile ranges) based on EMG onset and peak latencies. Participants gave written informed consent prior to data collection, and everyone received monetary compensation for their participation. The study was conducted in accordance with the Helsinki declaration, and was approved by the internal review board of the University of Oslo (ref. 1105078).

## Stop signal task and behavioral variables

All participants performed a choice-reaction time task and a stop signal task presented in alternating blocks (six blocks in total). Task presentation was controlled by E-prime 2.0 (Psychology Software Tools, Pittsburgh, PA). For this analysis, we focus on the stop signal task exclusively. The primary task was to respond with a button press using the thumb of the left or right hand to an arrow pointing to

the left or right, correspondingly. In 24% of the trials, a stop signal appeared after a variable SSD, instructing participants to stop their initiated response. In total, there were 684 go trials and 216 stop trials, equally distributed between left- and right-hand trials.

Each trial started with a black fixation cross (duration jittered between 500 and 1000 ms), followed by a colored arrow as a go signal, presented for 100 ms. In stop trials, a second arrow of a different color was presented after the SSD. The SSD could vary between 100 and 800 ms and was adjusted based on an adaptive tracking procedure collapsed across both hands: The initial SSD was set to 250 ms, and then increased or decreased by 50 ms following successful and unsuccessful stop trials, respectively. Responses were collected up to 1000 ms after the go stimulus and potential responses after this window were treated as omissions. All stimuli were presented centrally against a gray background and the stimulus color assignments (green, orange, or blue) were counterbalanced across participants. Trial order within a block was randomized, except for the first block, which always started with a minimum of 10 go trials. Short pauses (9 s) were given after every 75 trials, and a longer break with a duration of roughly 5 min was allowed halfway through the task.

Prior to the experiment, the participants completed a short training block (20 trials, 50% stop trials) with computerized trial-by-trial feedback. In addition, participants were told that it was important to be both fast and accurate, that they were not supposed to wait for the stop signal, and that it was not possible to successfully stop in all stop trials. Automated feedback was presented on screen after every 75th trial. Here, participants were instructed to be faster if their average goRT was above 600 ms, and to be more accurate if the average stop accuracy was below 40%. If neither of these applied, they received the feedback 'Well done!'.

The following behavioral variables were extracted based on task performance: go accuracies, error, and omission rates, go reaction times, stop accuracies, unsuccessful stop reaction times, average SSD, and SSRTs. These were calculated using the integration method, where corresponding go reaction time distributions included choice errors, and omissions were replaced by the maximum reaction time (*Verbruggen et al., 2019*).

We also estimated SSRTs and their variability using a Bayesian parametric approach (BEESTS; *Matzke et al., 2019*; *Matzke et al., 2013*). We chose the BEESTS2-GF model due to the low amount of choice errors in our sample. This model assumes a single go process as well as a stop process, and that both the go and SSRTs can be approximated by an ex-Gaussian distribution, defined by the $\mu$ and $\tau$ parameters and a standard deviation of $\sigma^2 + \tau^2$. As this model only fits one go process, choice errors were removed prior to analysis. We also removed outlier trials, defined as trials with RTs ± 3 SDs away from each participant's mean. Estimation was run using the Dynamic Models of Choice toolbox (*Heathcote et al., 2019*) in R Studio (version 1.3.1093) where models were fitted to each participant individually. Parameter priors were defined as weakly informative truncated normal distributions (*Heathcote et al., 2019*; *Skippen et al., 2019*). Specifically, we set the mean to 500 ms ($\mu_{go}$ and $\mu_{stop}$), 200 ms ($\sigma_{go}$ and $\tau_{go}$), and 100 ms ($\sigma_{stop}$ and $\tau_{stop}$), and the standard deviations to 1000ms. The priors for go and trigger failures had a mean of 1.5 (probit scale) with a standard deviation of 1. We ran 24 chains, 3 for each estimated parameter. All rhat-values were <1.1, indicating chain convergence. Posterior distributions were updated well, suggesting that priors were largely uninformative. Plots showing the prior relative to posterior distributions, chains, and posterior predictive model checks can be found in the accompanying materials on OSF (*Raud et al., 2021*).

## EMG acquisition

Two surface EMG electrodes were placed on the skin above the *abductor pollicis brevis* in parallel to the belly of the muscles on each hand using bipolar Ag/Ag-Cl montages. Corresponding ground electrodes were placed on each forearm. The recordings were performed using a BrainAmp ExG extension for bipolar recordings (Brain Products GmbH, Germany) with an online low-pass filter of 1000 Hz, a sampling rate of 5000 Hz, and a 0.5 µV resolution.

## EMG preprocessing and burst detection

EMG preprocessing and variable extractions were performed with custom scripts in MATLAB (The Math Works, Inc *MATLAB*, version 2018a) and eeglab (*Delorme and Makeig, 2004*; version 14.1.1b) and are publicly available on the OSF (*Raud et al., 2021*; https://osf.io/rqnuj/). The EMG signal was bandpass filtered between 20 and 250 Hz using a second-order butterworth filter and resampled to

500 Hz. Data epochs were extracted for all trials from −0.2 to 1.6 s relative to go signal onset. Trials were rejected at this stage if the average baseline activity was larger than 100 μV (across all participants, this led to the rejection of 206 trials, i.e., roughly 0.5 %).

The data epochs were transformed by taking the root mean square over a moving average window or ±5 data points, and normalized to baseline by dividing the full time-course by the average of the baseline period (−0.2 to 0 s relative to the go signal). For each participant and each hand, epochs were then concatenated and z-scored block-wise.

EMG bursts were identified in a trial if any data point exceeded a threshold of 1.2 (in standard deviation units, taking into account all trials). This threshold was confirmed visually and has been held constant across several recent studies (*Huster et al., 2020*; *Raud et al., 2020a*; *Raud et al., 2020b*; *Thunberg et al., 2020*). The peak latency was defined as the time-point of the highest amplitude in a trial. Onset latency was determined by tracking amplitude values backwards in time, starting at the peak latency, up until continuous data points of 8ms were below amplitude threshold. Lastly, peak latencies of the EMG in stop trials were recalculated relative to the stop signal onset by subtracting the SSD. All variables were averaged over left- and right-hand trials.

## Statistical analyses

Statistical analyses were performed using R (*R Core Team, 2019*) and the package collection of *tidyverse* (*Wickham et al., 2019*) and custom scripts in MATLAB. All figures were created either in MATLAB or in R using the packages *ggplot2* (*Wickham, 2020*), *ggpubr* (*Kassambara, 2020*), and *psych* (*Revelle, 2021*).

The trial differences between EMG amplitudes and peaks were tested by means of t-tests and analyses of variance within the Bayesian framework using the R package *BayesFactor* (*Morey et al., 2018*). The evidence was quantified using the Bayes factor, with values greater than 1 indicating support for the alternative and values below 1 for the null hypothesis (with >3 and <0.3 often considered indicators of strong evidence toward alternative or null, respectively).

We further tested whether the prEMG characteristics (inhibition functions, probability, onset latency, peak latency, and peak amplitude) varied across SSDs. To account for individual variability in SSDs, we centralized the SSDs per participant instead of using the absolute values. The mode of each participant's SSD distribution was treated as zero and we extracted the values ±3 SSD steps around the mode. The prEMG characteristics were averaged per SSD, so that each participant had one value per SSD, given that they had any trials with prEMG at that specific SSD.

The relationships between the prEMG characteristics and the prEMG were expected to be nonlinear with the exact shape unknown. Therefore, these were tested by the GAMMs using the R package *gamm4* (*Wood and Scheipl, 2020*). GAMM includes the possibility of adding a smooth term to the model, which is a linear sum of several splines. The degree of nonlinearity (or wiggliness) of the resulting model fit is determined by the edf and the corresponding F and p values, together with the estimated lambda factor that penalizes for overfitting. For example, in cases where linear fit is the best fit, the edf will be close to one; in case of second-order polynomial fit, the edf will be close to 2, etc. We considered smooth parameters with a p value below 0.05 as indicative of a significant effect.

Single-trial analysis. The single-trial analysis was done on prEMG (successful stop trials) only, using prEMG onset latencies, peak latencies, and AUC from onset to peaks. These were calculated as a cumulative sum of the amplitudes from onset to peak. AUC thus integrates the duration and amplitude of the prEMG bursts before their decline. As the bivariate correlations with the SSD at a single-trial level were high for the onset latencies ($r = 0.88$), they were orthogonalized for the SSD by predicting onsets from the SSD for each participant and using the residuals instead of original values. AUCs were log transformed since the original values were extremely skewed toward low values. Additional data cleaning was applied, in which trials were excluded if the peak latency occurred before the stop signal or if the onsets (go-locked) or peaks (stop-locked) exceeded two interquartile ranges of the individual participant's data. Three separate GAMMs were performed predicting peak latency from onset latency, AUC from peak latency, and AUC from onset latency.

Variability. Individual prEMG variability was estimated as the standard deviation of the prEMG peak latencies across trials for each participant. As some participants had implausibly large variability estimates, we identified and excluded four outliers who exceeded two interquartile ranges. One of these was also an outlier for the standard deviation of the SSRT, calculated by the BEESTS method. Bivariate

Pearson rank-ordered correlation coefficients and (uncorrected) p values were used to identify relationships between prEMG variability and the number of prEMG trials per participant, go reaction time standard deviations, average root mean square amplitudes during the intertrial interval of −20 to 0 ms before go stimulus, and standard deviation of the SSRT calculated by the BEESTS method. The scatterplots with the outliers included are in *Appendix 1—table 2*.

Reliability. We estimated the internal consistency of the different prEMG metrics (onset latencies, peak latencies, peak amplitudes, and AUCs from onset to peak) using a permutation-based split-half approach (*Parsons et al., 2019*). After the same level of additional data cleaning as in the single-trial analyses, we split the data for each participant into two random halves and calculated the mean of each half. These estimates for each half were then correlated across participants and the correlations were corrected using the Spearman-Brown prophecy formula (*Brown, 1910*; *Spearman, 1910*). This process was repeated 10,000 times, thus resulting in a distribution of reliability estimates. We used the mean of this distribution as our summary reliability estimate, and 2.5th and 97.5th percentiles as the upper and lower limits of the confidence intervals.

Initially, all trials from all participants were used for the reliability estimates. Next, we repeated the analysis iteratively using an equal number of trials per participant and excluding all participants who did not have enough trials available for a specific iteration. We started with 6 trials per participant (lowest number of trials available for all participants) and increased the number of trials per iteration by one until we were left with 32 trials per participant (with 12 participants remaining). We repeated this process 1000 times, thus getting a distribution of reliability estimates at each trial count.

## Meta-analysis

Three databases (Pubmed, Scopus, and Google Scholar) were searched with keywords ('EMG' OR 'electromyography') AND ('stop signal' OR 'countermanding' OR 'antisaccade' OR 'anticipated response') on 11/21/2021. The abstracts of 158 entries were screened and included in the meta-analysis if they reported prEMG peak or offset latencies relative to the stop signal onset. Most studies reported data from more than one experiment or experimental condition. All experiments/conditions were included as separate entries, except for *Thunberg et al., 2020*, from which we only included the three baseline sessions (and excluded the remaining six peri- and post-TDCS sessions, not to bias the results toward a single study with nine entries). Studies that reanalysed data already included in the meta-analysis were excluded. This resulted in 21 conditions/experiments from 10 different studies. In case of missing values, authors were contacted, and all authors provided the required values. We calculated the averages and standard deviations of the go RTs, SSRTs, prEMG detection frequencies, and peak/offset latencies, the differences between the SSRT and prEMG peak latencies, and correlations between the prEMG and SSRT. For the correlation coefficients, the means and standard deviations were calculated on Student z-transformed data, and then retransformed into correlation coefficients. The associations between the behavioral and EMG measures were tested with general mixed models with 'study' as a random variable, using the R package *lme4* (*Bates, 2022*). The spreadsheet with extracted data for all included studies is uploaded in the accompanying OSF page (*Raud et al., 2021*).

## Acknowledgements

We would like to thank Celina Müller, Sandra Klonteig, Josefine Bergseth, Thea Wiker, and Karl Zimmerman for their help with data collection. We further thank Ricci Hannah, Cheol Soh, Kelsey Sundby, and Jan Wessel for providing data for the meta-analysis.

## Additional information

### Funding

No external funding was received for this work.

## Author contributions
Liisa Raud, Conceptualization, Formal analysis, Investigation, Methodology, Software, Visualization, Writing - original draft, Writing - review and editing; Christina Thunberg, Conceptualization, Data curation, Formal analysis, Investigation, Methodology, Software, Writing - review and editing; René J Huster, Conceptualization, Investigation, Methodology, Software, Writing - review and editing

## Author ORCIDs
Liisa Raud http://orcid.org/0000-0003-2355-4308

## Ethics
Participants gave written informed consent prior to data collection, and everyone received monetary compensation for their participation. The study was conducted in accordance with the Helsinki declaration, and was approved by the internal review board of the University of Oslo (ref. 1105078).

## Decision letter and Author response
Decision letter https://doi.org/10.7554/eLife.70332.sa1
Author response https://doi.org/10.7554/eLife.70332.sa2

## Additional files

### Supplementary files
• Transparent reporting form

### Data availability
All data and analyses scripts are deposited in the Open Science Framework (Raud L, Thunberg C, Huster R. 2021. Partial response electromyography as a marker of action stopping. Data and analyses scripts. doi:https://doi.org/10.17605/OSF.IO/RQNUJ).

The following dataset was generated:

| Author(s) | Year | Dataset title | Dataset URL | Database and Identifier |
|---|---|---|---|---|
| Raud L, Thunberg C, Huster RJ | 2021 | Partial response electromyography as a marker of action stopping. Data and analyses scripts | https://osf.io/rqnuj/ | Open Science Framework, 10.17605/OSF.IO/RQNUJ |

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

## Appendix 1

### Replication analysis

Replication on publicly available dataset in https://osf.io/b2ng5/ from *Jana et al., 2020*.

$N$ = 41 (combined data from experiments 1 and 2). Results that divert from those reported in the original manuscript with discovery dataset are marked in **boldface** font.

### Behavior and EMG in the stop signal task

**Appendix 1—table 1.** Behavioral summary statistics.
RT = reaction time; sd = standard deviation; SSD = stop signal delay; SSRT = stop signal reaction time. $N$ = 41.

|  | Mean | sd | Median | min | max |
|---|---|---|---|---|---|
| Behavior |  |  |  |  |  |
| Go accuracy (%) | 97.95 | 2.26 | 98.44 | 86.67 | 100.00 |
| Go errors (%) | 0.66 | 1.07 | 0.22 | 0.00 | 5.11 |
| Go omissions (%) | 1.50 | 1.74 | 0.89 | 0.00 | 9.11 |
| Go RT (ms) | 481.28 | 80.33 | 471.67 | 335.16 | 636.69 |
| SSD (ms) | 271.18 | 93.04 | 255.17 | 88.33 | 453.21 |
| SSRT (ms) | 205.96 | 24.06 | 206.21 | 166.43 | 267.54 |
| Stop accuracy (%) | 50.00 | 2.09 | 50.00 | 42.67 | 54.00 |
| Unsuccessful stop RT (ms) | 441.05 | 72.64 | 436.43 | 318.15 | 591.36 |

**Appendix 1—table 2.** Means (and standard deviations in brackets) of the EMG variables.
Motor time refers to the difference between the EMG onset latency and the reaction time. Rise time refers to the duration of the rising flank of the EMG burst. *Peak latencies are time-locked to the stop signal onset. $N$ = 41.

| Variable | Go | Succ. stop | Unsucc. stop |
|---|---|---|---|
| Count | 430 (15) | 39 (11) | 74 (3) |
| Percentage | 99 (1) | 58 (17) | 99 (2) |
| Onset latency (ms) | 367 (81) | 369 (85) | 316 (67) |
| Peak latency* (ms) | NA (NA) | 140 (27) | 74 (36) |
| Peak amplitude ($z$) | 11 (1) | 6 (1) | 10 (1) |
| Motor time (ms) | 118 (16) | NA (NA) | 118 (16) |
| Rise time (ms) | 60 (12) | 30 (7) | 57 (11) |
| AUC ($z$) | 138 (14) | 51 (12) | 124 (15) |

### Statistical results

EMG onset latencies:

- unsuccessful stop < go (BF = $1.26 \times 10^{16}$)
- prEMG ~ go (BF = 0.21)

EMG peak amplitudes:

- successful stop < go (BF = $1.30 \times 10^{22}$)
- successful stop < unsuccessful stop (BF = $5.60 \times 10^{16}$)
- unsuccessful stop < go (BF = 13,780)

Motor time:

- **unsuccessful stop ~ go (BF = 0.17)**

Rise time:
- unsuccessful stop < go (BF = 393,283)

## Associations of prEMG latency and frequency with behavior
- correlation SSRT – goRT ($r = -0.55$, p < 0.001)
- correlation goRT – prEMG latency ($r = -0.52$, p < 0.001)
- correlation goRT – prEMG frequency ($r = -0.39$, p = 0.01)
- correlation prEMG frequency – latency ($r = 0.01$, p = 0.935)
- correlation prEMG frequency – SSRT ($r = 0.21$, p = 0.167)
- correlation SSRT-prEMG latency ($r = 0.56$, p < 0.001)

## Comparison of behavior- and EMG-based inhibition functions
GAMM: probability of response as a function of SSD and response type (button press/EMG).

Model $R^2$ = 0.75.

Significant effect of SSD (df = 3.30, $F$ = 281.95, p < $2 \times 10^{-16}$); response type ($b$ = 0.07, $t$ = 9.52, p < $2 \times 10^{-16}$), and interaction between SSD and response type (df = 2.21, $F$ = 29.03, p = $2 \times 10^{-16}$).

## Context independence of going and stopping
Statistical tests for RTs:
- unsuccessful stop < go (BF = $9.57 \times 10^{13}$)
- unsuccessful stop > fast go (using means BF = $1.3 \times 10^9$, medians BF = 64,049)

Statistical tests for EMG onsets:
- unsuccessful > fast go (BF = $2.96 \times 10^9$)
- prEMG < medium go (BF = 34,849)

**Appendix 1—table 3.** EMG onsets in go trials divided into fast (between go onset and p(press|stop)), medium (between p(press|stop), and p(EMG|stop)) and slow (>p(EMG|stop)) trials. usEMG = unsuccessful stop EMG; prEMG = partial response EMG (successful stop trials).

|  | Mean | sd | Median | min | max |
|---|---|---|---|---|---|
| goEMG_fast | 411 | 82 | 412 | 260 | 678 |
| goEMG_medium | 520 | 103 | 503 | 345 | 858 |
| goEMG_slow | 626 | 139 | 584 | 415 | 1,068 |
| usEMG | 431 | 87 | 433 | 272 | 704 |
| prEMG | 496 | 114 | 479 | 290 | 841 |

## prEMG characteristics as a function of SSD
Four GAMMs to predict the prEMG probability, onsets, amplitudes, and peak latencies from the SSDs:
- probability: df = 2.42, $F$ = 42.88, p < $0.01 \times 10^{-15}$, $R^2$ = 0.25
- onset latency: df = 1.00, $F$ = 655.40, p < $0.01 \times 10^{-15}$, $R^2$ = 0.36
- peak latency: df = 1.00, $F$ = 19.24, p = $2 \times 10^{-5}$, $R^2$ = 0.06
- peak amplitude: df = 1.00, $F$ = 12.52, p = $5 \times 10^{-5}$, $R^2$ = 0.04

## prEMG peak latency as a function of time on task
GAMM model predicting prEMG peak latency by trial order while controlling for SSD: df = 1.73, $F$ = 0.56, p = 0.442.

GAMM model predicting probability of producing prEMG by trial order while controlling for SSD: df = 2.38, $\chi^2$ = 5.97, p = 0.233.

## Single-trial associations of prEMG onset latency, peak latency, and AUC
GAMMs predicting: (1) peak latency from onset latency: edf = 3.36, $F$ = 1119, p < $2 \times 10^{-16}$, $R^2$ = 0.57

1. peak latency from AUC: edf = 1.00, $F$ = 12.96, p = 0.0003, $R^2$ < 0.001
2. onset latency from AUC: **edf = 2.59**, $F$ = 8.67, p < $2 \times 10^{-16}$, $R^2$ = 0.12

## prEMG latency variability

The average individual standard deviation was 45ms (group-level sd = 20) and varied from 16 to 87 ms.

## prEMG split-half reliability

**Appendix 1—table 4.** Split-half reliability estimates for partial response electromyography (prEMG) characteristics, including mean, standard deviation, and percentiles derived from the permutation distributions that correspond to the 95% confidence intervals.

RMS = root mean square; AUC = area under the curve from onset to peak.

|  | Mean | sd | 2.5th | 97.5th |
|---|---|---|---|---|
| Onset latency (ms) | 0.98 | 0.01 | 0.96 | 0.99 |
| Peak latency (ms) | 0.92 | 0.05 | 0.91 | 0.96 |
| Peak RMS amplitude (z) | 0.59 | 0.11 | 0.32 | 0.77 |
| AUC (z) | 0.53 | 0.12 | 0.27 | 0.73 |

### Inhibition functions

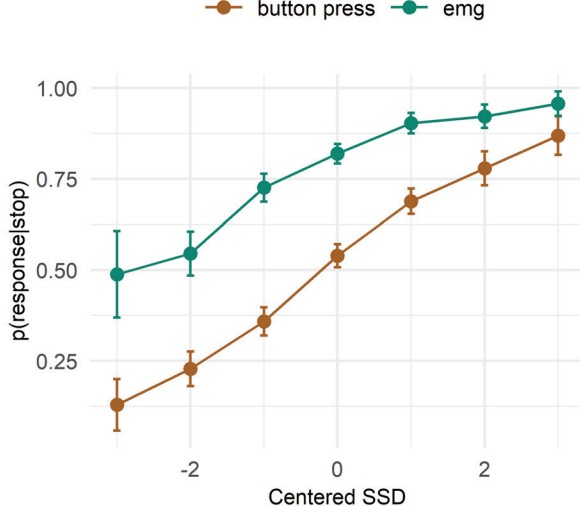

**Appendix 1—figure 1.** Inhibition functions. Error bars represent 95% confidence intervals. Note that the error bars are calculated across all trials and participants, ignoring that each participant has several data points.

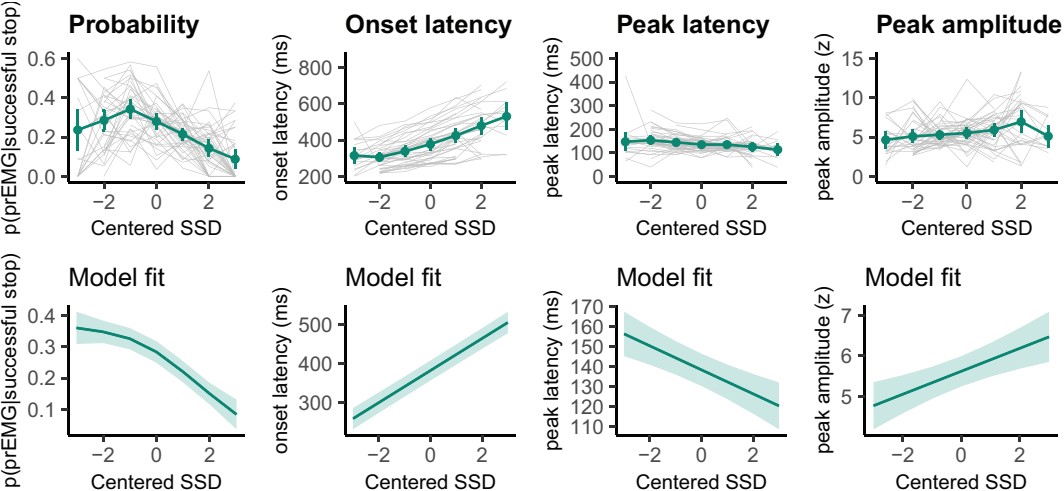

**Appendix 1—figure 2.** Partial response electromyography (prEMG) probability, onset latency, peak latency, and peak amplitude. The upper panels represent empirical data across all participants. Green lines represent the group average with 95% confidence intervals and gray lines represent each participant. The lower panels depict the model fit of the generalized additive models with 95% confidence intervals. Note the different *y*-axes between empirical data and model fits.

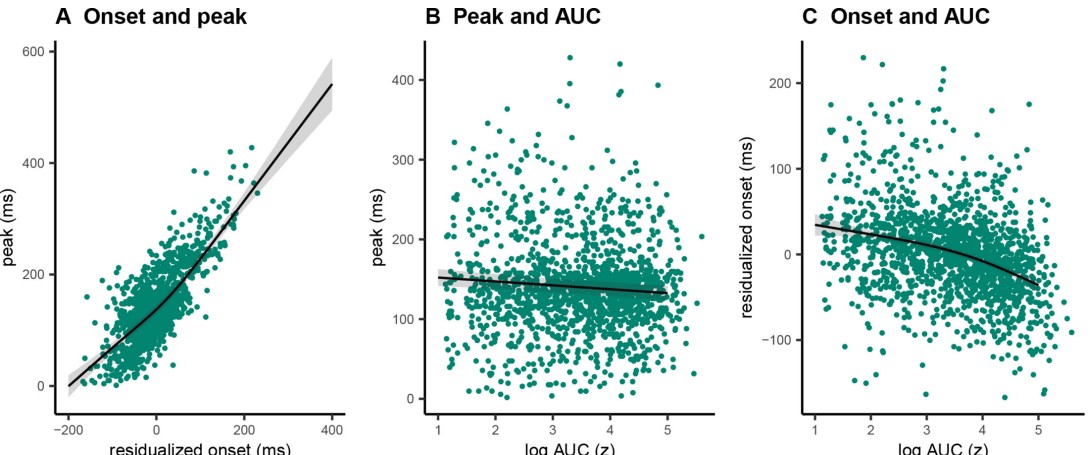

**Appendix 1—figure 3.** Single-trial associations between partial response electromyography (prEMG) onset latency, peak latency, and area under curve (AUC). The dots represent single trials from all participants. The black lines represent the fixed effect model fits from generalized additive mixed models, where the individual variability was modeled as separate intercepts for each participant. The shaded areas represent 95% confidence intervals.

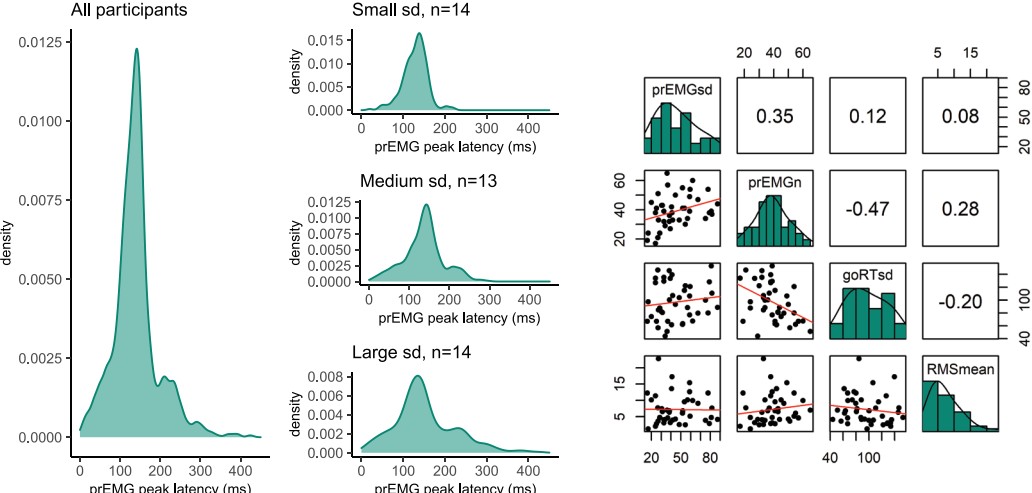

**Appendix 1—figure 4.** Variability of partial response electromyography (prEMG) peak latency. Left: density functions of prEMG peak latency across all trials and participants (left panel) and separately for group of participants with low, medium, and large individual standard deviations (sd; right panels). Right: correlation matrix between individual prEMG standard deviations (prEMGsd), number of prEMG trials (prEMGn), go reaction time standard deviations (goRTsd), average root mean square EMG amplitudes during the intertrial interval −200 to 0 before go stimulus onset (RMSmean). The diagonal shows the histograms for each variable together with the density functions, the lower diagonal shows the scatterplots, and the upper diagonal shows the Spearman correlation coefficients.

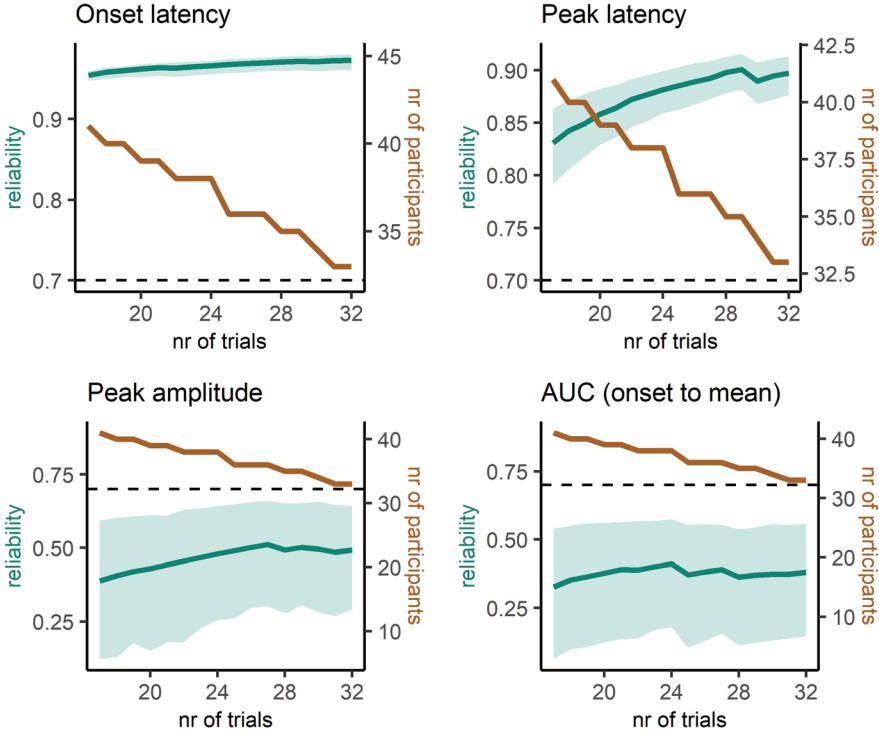

**Appendix 1—figure 5.** Reliability estimates as a function of number of trials (green) and number of participants (brown) for partial response electromyography (prEMG) latency- and amplitude-based measures. Green shaded areas represent 95% confidence intervals for the reliability estimates, and the black dashed horizontal line represent a reliability value of 0.7 for reference. Note the different *y*-axes for latency and amplitude measures. AUC = area under the curve from prEMG onset to peak.

## Appendix 2

## Scatterplots of prEMG latency variability with outliers

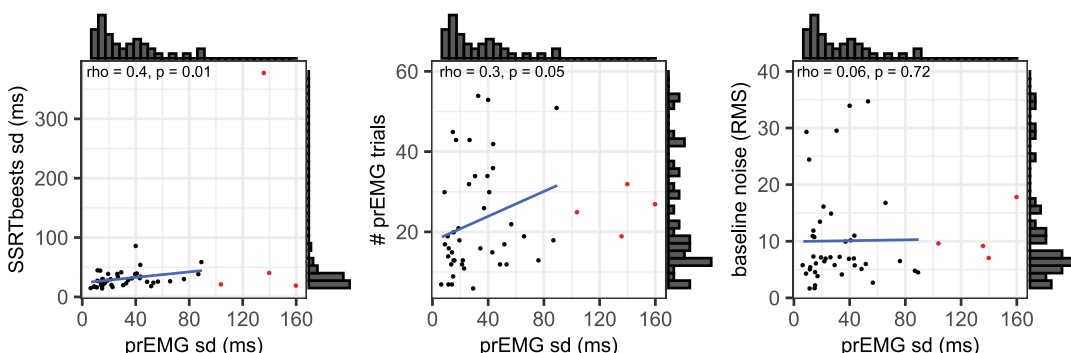

**Appendix 2—figure 1.** Scatterplots and histograms for the associations of the partial response electromyography (prEMG) peak latency standard deviations with stop signal reaction time (SSRT) standard deviations derived from BEESTS (left panel), number of prEMG trials (middle panel), and baseline EMG noise (right panel). The red dots represent outliers, while the histograms, rho/p values, and blue regression lines are calculated after excluding the outliers.

