## [Editor Report]

The authors propose that the covert latency of the stopping process, which allows for the stopping of movements in tasks like the stop-signal paradigm, can be measured through the offset latency of bursts of electromyographic (EMG) activity that is observable on trials in which no overt response (typically a button press) was produced. The investigation is extensive and systematic and will provide a helpful methodological resource that rigorously specifies this alternative measure of stopping.

---

## [Decision Letter]

**Decision letter after peer review:**

[Editors’ note: the authors submitted for reconsideration following the decision after peer review. What follows is the decision letter after the first round of review.]

Thank you for submitting the paper "Partial response electromyography as a marker of the individual stopping latency" for consideration by *eLife*. Your article has been reviewed by 2 peer reviewers, and the evaluation has been overseen by a Reviewing Editor and a Senior Editor. The following individual involved in review of your submission has agreed to reveal their identity: Inge Leunissen (Reviewer #2).

Comments to the Authors:

We are sorry to say that, after consultation with the reviewers, we have decided that this work will not be considered further for publication by *eLife*.

Both reviewers found the work to be important and timely, given the growing use of pmEMG as a measure of stopping. However, in review and discussion, it was not clear to all reviewers that the data fully support the conclusion that prEMG is an unambiguous marker of stopping latency. Though one reviewer was more positive in their review, both reviewers highlighted concerns that were potential serious. For example, there was agreement that only a small percentage of trials show the bursting effect. This raised concern in consultation about the conclusions. Indeed, one reviewer commented that these results could have led to the opposite conclusion, namely that much is still unknown about prEMG and its relationship to stopping, and so it is premature to declare it a viable marker of stopping latency. In light of these ambiguities, it was decided that the paper was not suitable for *eLife*. I have included the full reviewer comments appended below to convey the basis for this decision.

*Reviewer #1:*

The current article proposes the usage of "partial response electromyography (prEMG)" to measure inhibitory control processing in the context of the stop-signal task. Specifically, the idea is to use single-trial EMG to detect subthreshold activity (and its subsequent reduction) on trials in which no overt response was made (i.e., signal-inhibit or "successful stop" trials). According to the authors' proposal, this would allow for a single-trial quantification of the latency of the stopping process.

This possibility is attractive for obvious reasons, chiefly the fact that the latency of the stopping process in the stop-signal paradigm is not directly or overtly observable from behavioral data alone. While this "partial response EMG" approach has been used in several prominent papers in the 90s, it has recently seen a resurgence, largely due to prior work by the current group of investigators, as well as others (notably including a paper from the Aron group in this journal). This resurgence is largely due to the explicit proposal that the offset latency of the prEMG in particular could be used as a proxy of the latency of the stopping process on individual trials.

Here, the authors perform an extensive investigation of several psychometric properties of prEMG as well as its relationship to behavior and other aspects of the stop-signal task.

Based on this, the authors propose that prEMG (and in particular, its offset latency) is a "unique and reliable measure of stopping" and "encourage the widespread us of prEMG to investigate the underlying mechanisms of response inhibition".

Interestingly, based on the results of their current investigation, which is thorough and extensive in many ways, I actually arrive at a different conclusion.

Indeed, while I completely agree that there is utility to prEMG as a measurement, I think there are many theoretical and practical considerations that severely limit that utility. Furthermore, while the authors may turn out to be right that prEMG is a "reliable measure of stopping", I am quite unconvinced that this has been conclusively shown yet, including in the current paper. Indeed, some of the data presented here casts even more doubt on that purported association.

Chiefly, my concerns are as follows:

– First, I see no definitive reason to assume that the reduction of the EMG that constitutes the prEMG measurement is actually due to an inhibitory process. While this may well be the case, it also seems entirely possible that the observed reduction of the EMG is merely due to a cessation of the invigoration of the go-response by the motor system, which could happen entirely in the absence of inhibitory processing. Furthermore, alternatively, it could also reflect an initial subthreshold activation of the go response in those trials. At this point, as much as the authors (and others, including myself) may prefer it, there seems to be little evidence to favor the inhibition-based interpretation over the others – at least to me.

– Second, as the authors themselves note (to their credit), the correlation between prEMG and SSRT is only very moderate – as low as ~10% variance explained, depending on the study. I understand and agree that SSRT has problems as a measurement, but it is also true that relationships to SSRT have been used as the ultimate benchmark when evaluating the utility of a physiological measurement as a potential "biomarker" of stopping – including by the current authors at several points of their manuscript. At face value, the low correlation to SSRT further casts doubt on the proposed association between prEMG and inhibition / stopping behavior.

– Third, while again I agree with the authors that SSRT has been the subject of recent (justified) criticism, I am not aware of the fact that anybody is arguing for abolishing it altogether. Of course, work by investigators like Matzke, Bissett, Verbruggen and others have outlined boundary conditions under which SSRT estimates can become imprecise (though corrections exist in most cases). But the same is of course true of the EMG measurement, which is subject to considerable noise, as well as crosstalk from other muscles – including, notably, the antagonist of the responding muscle, etc. Hence, it is at least doubtful to me that prEMG is inherently superior in regard to its associated principled problems (some of which I will get into further below).

– Fourth, this leaves the single-trial argument, which I agree is quite compelling: SSRT is a compound measure, whereas prEMG carries some degree of single-trial information (though compared to the compound mean-prEMG measure, its single-trial reliability is more limited, as shown in the current study). However, this glosses over the highly pertinent aspect that only a remarkably low fraction of successful stop-trials even shows a prEMG burst (23% in the current study!). As such, there is a huge problem in that trials on which prEMG can be measured do not represent the typical stop-signal trial (or even their majority), and instead only reflect a very minor portion of the trials. The proposition that we can learn something general and universal about the behavior in a certain task by only looking at a small fraction of atypical trials in the critical condition seems like a huge deal to me, which is not really reflected in the considerations presented in this article at all. This problem is amplified by the fact that I imagine that there is some degree of dispersion around that mean, meaning that there will be subjects in which the fraction of prEMG trials is even lower than this 23%. How much are we really gaining in terms of an 'overt' measure of the stopping process if it is only present in such a small subset of trials?

– Fifth, the trial-wise variation of prEMG latency is quite considerable. While the authors argue that there is some informational value in that intra-subject standard deviation (which may well be accurate, though the association with other subject variables is quite moderate at best; in fact, the highest correlation found for this measurement was to the number of trials, rather than some meaningful behavioral measure or task variable), I find it very hard to believe that the true stopping process within a given subject varies by as much as a standard deviation as large as 143ms would indicate. Indeed, the BEESTS method estimates the SD of SSRT to be <50ms at the largest, as is evident from Figure 7B. The comparably enormous variation of the prEMG measure seems therefore underappreciated.

– Sixth, how does it make sense that prEMG occurs most prominently at short SSDs? Intuitively, wouldn't one presume that prEMG is most likely to occur when stopping happens "at the last second" – i.e., after cortico-spinal volleys have already been fired and the responding muscle has begun to contract? If so, shouldn't that be _least_ likely when SSDs are short?

– Seventh, the authors cite the recent Tatz et al. paper as an independent replication of the association between prEMG and SSRT. However, they never discuss what seems to be the core thrust of that study, which is that reductions of EMG are not specific to stopping and appear to occur after other events as well.

– Finally, the theoretical framing of the paper seems a bit unclear to me. The authors initially frame prEMG as an "alternative to SSRT" and "a more direct measure of the stopping latency" (p. 3f). However, they then argue that prEMG and SSRT are "partially independent" (which, incidentally, could be better phrased – measures are either independent or they are not). In the Conclusions, the authors then unequivocally state that "prEMG is a unique and reliable measure of stopping". So, if I understand the take home message correctly, the authors suggest that prEMG is a measure of stopping, which is (partially?) independent from SSRT, which, however, is also a "behavioral measure" of stopping (p. 23). In other words, the proposed relationship between prEMG, SSRT, and stopping should be more explicit. Do they both measure stopping? If so, how is that sensible? Or does only one of them? Should we discard SSRT in favor of prEMG? If the latter is to be the argument, I would put forward the above concerns, and reiterate that an explicit proof that prEMG reflects an inhibitory effort is still outstanding. If both SSRT and prEMG measure stopping, then how can they be only very moderately correlated and occur on different time scales?

In summary, I remain rather unconvinced that prEMG is as of yet established as an "unambiguous marker of the stopping latency", which is the study's primary claim. It is also not clear to me if the additional set of problems associated with this measure makes it a viable "alternative to SSRT", as is implied. Hence, while I appreciate that two flawed measures (SSRT and prEMG) may be superior to only one, even if their relationship is unclear, I still believe it's too early to "encourage the widespread us of prEMG to investigate the underlying mechanisms of response inhibition".

*Reviewer #2:*

Raud et al. explore the possibility of using partial electromyography responses on successful stop trials as a physiological measure to capture individual stopping latencies (prEMG). They provide an extensive overview of the measure in a large stop signal task data set, how it compares to traditional measures, how it relates to the assumptions underlying the horse-race model and its variability / reliability. In addition, it provides practical tips on how to collect the EMG data and to extract the partial responses. Overall, the data analysis is rigorous, and the conclusions are justified by the data. The main limitation of the approach is that partial responses were only detected on 23% of all successful stop trials. Previous literature has reported incidence of ~50%, but it remains a question whether this portion of the data is representative of all stopping behavior. The conclusion that the prEMG does not support the independent horse race model could have severe implications. It is not the first account of violations of the independence assumption of the model, which is widely used for the estimation of inhibition speed. It is vital to investigate the degree of violation in both response times and partial EMG responses under different task conditions to get an idea of the degree of the violation and its impact on the stopping latency estimations. I commend the efforts of the authors and would very much like the field to adopt the inclusion of these EMG recordings.

1. The percentage of successful stops with a partial EMG response seems to be quite low (~23%). In my own experience the percentage was more around 40 (with a choice reaction time version of the SST) to 50% (with an anticipated response version of the SST) (data associated with Leunissen et al. 2017 Eur J Neuroscience, unpublished). Also in the paper of Coxon et al. 2006 and Jana et al. 2020 the percentage seems to be quite a bit higher (for the latter even 67%). I wonder whether the focus on the go task (and not slowing down here) can explain these differences. The RTs reported in your experiment are fairly long. Providing more feedback on the go task might help to increase the percentage of successful stops with a partial response. This would be important for at least two reasons: it is unclear how well 23% of all successful stops represent stopping behavior. Moreover, most researchers do not present 216 stop trials, but rather only 80-100. Leaving one with only ~10 partial responses per participant. The willingness to adopt such a lengthy protocol in addition to the EMG might be low, therefore I think it would be good to reflect on this.

2. APB vs FDI: You report the partial EMG responses from a downwards thumb movement in the APB. Of course, this makes sense for this movement direction, but in my experience it is much easier to get a good signal to noise ratio in the EMG for the FDI (which would require and abduction of the index finger or a different hand position). See Coxon et al. 2006 J Neurophysiology Figure 1 and 3 for a comparison of the EMG traces of the APB and FDI. Having a better SNR might also help in identifying the partial responses in a greater number of trials.

3. You show a dependency between the go and stop process based on the EMG analyses. It might be of added value to check the level of dependency in the response times as well. For example based on the BEEST approach outlined by Dora Matzke in this pre-print (https://psyarxiv.com/9h3v7/). It would be interesting to see if there is agreement between the two approached with respect to the (lack) of independence. It would also be nice to be able to compare the RTs on unsuccessful stops and the fast go-RT portion (discussed on page 14, line 215 but not shown).

The conclusion that there is dependency between the go and stop process seems to hinge on the finding that the average EMG onset in unsuccessful stop trials was slower than the average EMG onset in fast go trials. Could a difference in power perhaps produce this result? You have many more go trials than unsuccessful stop trials, would the full distribution simply not have been sampled in the unsuccessful stop trials?

4. In my opinion the discussion could state a bit more clearly what the benefit of using the EMG peak latency on successful stop trials is compared to a parametric approach of analyzing the response times for example. Should it really be EMG or could one also simply use a force censor to obtain a similar measure? What are possible disadvantages of EMG? For example, in clinical populations one will find that the background EMG is much higher, making it more difficult to identify the partial responses.

5. I think it would be helpful to include a visualization of the stop signal task used.

6. Please include a figure that visualizes how the different latencies are determined.

7. Rise time is defined as EMG onset to peak latency. But peak latency was the time between the stop signal and the peak amplitude in the EMG, obviously there is no peak latency in go trials. Please redefine for clarity.

8. Motor time was longer in unsuccessful stop trials than in go trials. Was this true for all portions of go trials in the distribution? Response force on go trials tends to increase with the increasing likely hood of a stop signal appearing (i.e. participants wait and then respond very quickly and forcefully) (van den Wildenberg et al. 2003 Acta Psychol). Those trials might have a higher RT but a very short motor time.

9. Could the average areas under the curve be added to table 2? This would be helpful for understanding the analysis on the different dynamics of going and stopping.

10. Page 24, line 370-371. Comment regarding SSRT being associated with a speed-accuracy trade off suggests that this is (possibly) not the case for prEMG. Give that this speed-accuracy tradeoff reflects the strategic choices of the participant I assume that this should also be reflected in the (number, size, ...) of the prEMG.

---

## [Author Response]

[Editors’ note: The authors appealed the original decision. What follows is the authors’ response to the first round of review.]

Reviewer #1:[…]Chiefly, my concerns are as follows:– First, I see no definitive reason to assume that the reduction of the EMG that constitutes the prEMG measurement is actually due to an inhibitory process. While this may well be the case, it also seems entirely possible that the observed reduction of the EMG is merely due to a cessation of the invigoration of the go-response by the motor system, which could happen entirely in the absence of inhibitory processing. Furthermore, alternatively, it could also reflect an initial subthreshold activation of the go response in those trials. At this point, as much as the authors (and others, including myself) may prefer it, there seems to be little evidence to favor the inhibition-based interpretation over the others – at least to me.

We agree with this point. Importantly, it is not our goal to ‘sell’ prEMG as an unambiguous marker of response inhibition – we agree with the reviewer that there is no evidence for this. Yet, by definition, the decline of prEMG is a marker of stopping already initiated actions. Correspondingly, we recognize prEMG as an extremely useful tool, as it allows us to address whether action stopping is achieved by an active inhibitory process or other mechanisms in the brain. This is what motivated us to submit the manuscript in the ‘tools and methods’ section of the journal, not as a separate empirical finding.

We had addressed this point the original submission’s discussion in a section entitled ‘Outstanding question 1: Is prEMG peak latency a marker of individual inhibition capacity. Given that we changed the outline of the discussion, this point is now discussed in paragraph ‘Response inhibition and prEMG’ in the revised manuscript. As this point is critical to the interpretation of the prEMG, we have further elaborated on it in the introduction (lines 87-93), changed the title and abstract correspondingly, and screened the manuscript carefully for ambiguities in wording, so that in no place we would indicate that prEMG decline is a biomarker of inhibition as such.

– Second, as the authors themselves note (to their credit), the correlation between prEMG and SSRT is only very moderate – as low as ~10% variance explained, depending on the study. I understand and agree that SSRT has problems as a measurement, but it is also true that relationships to SSRT have been used as the ultimate benchmark when evaluating the utility of a physiological measurement as a potential "biomarker" of stopping – including by the current authors at several points of their manuscript. At face value, the low correlation to SSRT further casts doubt on the proposed association between prEMG and inhibition / stopping behavior.

The associations between the SSRT and the prEMG vary greatly in the studies, which is now communicated better via the added meta-analysis in which we found a mean effect size of r=0.64 (lines 188-192). Interestingly, most individual studies consider this correlation as evidence for the prEMG latency and the SSRT to capture the same process, but at different time-points (at stages in the peripheral nervous system and at the behavioral level, respectively). The time-difference between the two also varies greatly between the studies but is typically ‘explained away’ by the electromechanical delay, that is the delay between muscle contraction/relaxation and behavioral response.

We agree with the reviewer that this across-study variability indicates weak evidence that the SSRT and the prEMG latency captures the same underlying process. However, we disagree that this somehow invalidates the prEMG measure.

The fact that the SSRT has been used as a benchmark stopping-measure is of course true, but it does not mean that one should dismiss new evidence suggesting that this approach is not as straight-forwards as it was believed to be. We find it counter-intuitive that a new measure should be rejected on the grounds that it does not fully correspond to an existing flawed measure. We have now added a paragraph directly discussing the interpretation of the prEMG and the SSRT (lines 431-465) and listed the advantages and disadvantages of the SSRT and prEMG in the conclusion (lines 552-561).

– Third, while again I agree with the authors that SSRT has been the subject of recent (justified) criticism, I am not aware of the fact that anybody is arguing for abolishing it altogether. Of course, work by investigators like Matzke, Bissett, Verbruggen and others have outlined boundary conditions under which SSRT estimates can become imprecise (though corrections exist in most cases). But the same is of course true of the EMG measurement, which is subject to considerable noise, as well as crosstalk from other muscles – including, notably, the antagonist of the responding muscle, etc. Hence, it is at least doubtful to me that prEMG is inherently superior in regard to its associated principled problems (some of which I will get into further below).

Abolishing the SSRT and replacing it with the prEMG is not our goal. We also do not propose that prEMG is inherently superior to the SSRT, precisely because we think they capture complementary aspects of stopping behavior. However, we do propose to re-evaluate the interpretation of the SSRT, as it has meanwhile been shown that it is not ‘un-ambiguous’ marker of stopping either (as the work by Matzke, Bisset et al. cited by the reviewer indicates). Whereas it is true that physiological raw signals are noisy, prEMG’s split-half reliability of 0.87 (and 0.92 in the replication dataset) for its peak latency showcases excellent reliability.

– Fourth, this leaves the single-trial argument, which I agree is quite compelling: SSRT is a compound measure, whereas prEMG carries some degree of single-trial information (though compared to the compound mean-prEMG measure, its single-trial reliability is more limited, as shown in the current study). However, this glosses over the highly pertinent aspect that only a remarkably low fraction of successful stop-trials even shows a prEMG burst (23% in the current study!). As such, there is a huge problem in that trials on which prEMG can be measured do not represent the typical stop-signal trial (or even their majority), and instead only reflect a very minor portion of the trials. The proposition that we can learn something general and universal about the behavior in a certain task by only looking at a small fraction of atypical trials in the critical condition seems like a huge deal to me, which is not really reflected in the considerations presented in this article at all. This problem is amplified by the fact that I imagine that there is some degree of dispersion around that mean, meaning that there will be subjects in which the fraction of prEMG trials is even lower than this 23%. How much are we really gaining in terms of an 'overt' measure of the stopping process if it is only present in such a small subset of trials?

We approach this issue with an empirical and a theoretical argument.

Starting with the empirical: based on our experience with different response devices, we suspect that the low number of prEMG trials in this study was caused by a low resistance of the response device buttons, so that even minor thumb movements resulted in a button press. This may limit the generalizability of prEMG across studies, so we performed several extra analyses to gauge the extent of this issue.

1) First, we looked at the associations of the prEMG frequency with the goRT, SSRT, and the prEMG peak latency across participants in the current study (lines 162-177). We found a moderate negative correlation with the goRT, indicating that faster participants had more prEMG trials. The correlations of the prEMG detection frequency with the prEMG peak latency and SSRT were small and not significant. This is in context of a strong negative correlation between the goRT and the SSRT. As such, there is evidence for trade-off between fast going and slow stopping. This relationship seems to some degree affect prEMG frequency, but less so its latency, which supports the usability of the prEMG latency as a measure of action stopping.

2) Second, we quantified the same associations across studies via a meta-analysis (lines 178-194, 706-723, Figure 2). The strong negative association between goRT and the SSRT prevailed also across studies, as did the relationship between goRT and prEMG frequency. The prEMG peak latencies were relatively stable across different experiments and were not associated with prEMG detection frequency.

3) Third, this still leaves us with the question whether the psychometric properties of the prEMG, as quantified in the current sample, generalize to a sample with more prEMG trials. We therefore applied our pipeline to a previously published dataset from a different lab with ~60% of stop trials with prEMG (Jana et al. 2020 experiments 1 and 2; data available at https://osf.io/b2ng5/). Most of the results replicated and many of the trends were even more clearly visible in the replication dataset. The few instances that did not replicate are now reported in the manuscript (lines 149-161; 342-343). The full replication results are available in Appendix 1.

Now to the theoretical argument. The reviewer argues that the prEMG potentially does not reflect a ‘typical stop trial’. This implies that a ‘typical stop trial’ exists in the first place, reflecting the long tradition of conceptualizing stopping in the horse race model framework. The reviewer agrees that the horse race model assumptions are a limiting factor for the SSRT and mentions that corrections for the violation of its underlying assumptions exist for most cases. However, these correction methods rely on the removal of participants and/or trials that do not fit said assumptions, or the replacement of such non-typical trials based on further assumptions.

This inherently implies that there exists a ‘typical stop trial’ which uses the ‘correct’ inhibition process, while trials/participants that deviate from an expected pattern are simply discarded without any further explanation. Notably, this limits the applicability of the SSRT as well as its interpretation, as violations of the assumptions of the horse race model may be common in-patient groups.

With EMG, we can quantify the variability of stopping latencies across trials, impose trial classifications to address potential sub-processes of action stopping (e.g. by dissociating trials with and without prEMG, or trials in which EMG started before or after the stop signal etc.) and complement such analyses with other physiological measurements.

In addition, more elaborate task designs can be used to acquire even more information, e.g. on antagonist muscle movements (Atsma et al., 2018; Goonetilleke et al., 2010), partial errors in compound movements (Goonetilleke et al., 2012), or the contraction and relaxation times of different muscle groups (Havas et al., 2020). In sum, whether there exists a ‘typical stop-trial’ is an empirical question, and it is one that the prEMG can provide answers to.

We have now included a comprehensive discussion on the generalizability of the prEMG in the discussion (lines 467-524).

– Fifth, the trial-wise variation of prEMG latency is quite considerable. While the authors argue that there is some informational value in that intra-subject standard deviation (which may well be accurate, though the association with other subject variables is quite moderate at best; in fact, the highest correlation found for this measurement was to the number of trials, rather than some meaningful behavioral measure or task variable), I find it very hard to believe that the true stopping process within a given subject varies by as much as a standard deviation as large as 143ms would indicate. Indeed, the BEESTS method estimates the SD of SSRT to be <50ms at the largest, as is evident from Figure 7B. The comparably enormous variation of the prEMG measure seems therefore underappreciated.

The standard deviations of prEMG are right-skewed with a few participants having very large values. Given that we used Spearman’s rank-ordered correlations for the variability analysis, we were not concerned about the outliers affecting our results. In hindsight, this was not the best choice, as it indeed gives an impression of implausibly large variability across participants. We have now excluded 4 outliers who exceeded two inter-quartile ranges from the mean. (Note that one of these was already omitted in the original submission due to implausibly large standard deviation of the BEESTS-SSRT). The corrected standard deviation is now 31ms (ranging from 7-90 ms; changes in Table 1, Figure 8, lines 356-357, 370-375, and 683-690).

Omitting these outliers did not have major impact on the correlation analysis (see Author response table 1). We additionally repeated the analysis including only participants with at least 10 prEMG trials (n=37), again without any major differences in the results.

**Author response table 1. sa2table1:** 

Relationship between prEMG standard deviation and …	Original analysis	Revised analysis after outlier removal
number of prEMG trials	0.34	0.30
baseline EMG (noise)	0.13	0.06
go RT standard deviation	0.18	-0.01
BEESTS-SSRT standard deviation	0.33	0.40

The scatter plots for the prEMG variability data can be found in Appendix 2. We overlayed the outliers in red to illustrate the potential effects on the results.

– Sixth, how does it make sense that prEMG occurs most prominently at short SSDs? Intuitively, wouldn't one presume that prEMG is most likely to occur when stopping happens "at the last second" – i.e., after cortico-spinal volleys have already been fired and the responding muscle has begun to contract? If so, shouldn't that be _least_ likely when SSDs are short?

This is an alternative hypothesis, which would indicate dependence of the stopping process on the activation of cortico-spinal volleys, which occurs more often at longer SSDs. In contrast, our hypothesis, based on previous literature (Coxon et al., 2006), was that stop process is triggered by the stop signal independently of the status of the cortico-spinal volleys. Instead, the decreased prEMG frequency at higher SSDs would result from the fact that it is more difficult to stop a response with a high SSD, thus many of these trials end up being classified as unsuccessful stop trials instead. This is indeed supported by our data.

– Seventh, the authors cite the recent Tatz et al. paper as an independent replication of the association between prEMG and SSRT. However, they never discuss what seems to be the core thrust of that study, which is that reductions of EMG are not specific to stopping and appear to occur after other events as well.

The paper by Tatz et al. (2021) is an excellent demonstration of the usefulness of the EMG approach to study the neural mechanisms of inhibitory control. In the authors words “the current results are the first demonstration that this [prEMG] signature is not uniquely indicative of inhibitory control during action-stopping, but again common to all salient events, just like the CSE suppression. This is further in line with the assertion that the rapid invocation of inhibitory control is a stereotypic consequence of stimulus-driven attentional capture (Wessel and Aron, 2017).” We have now referenced this article as a good example for investigating mechanisms of cognitive control using EMG in the manuscript lines 521-523.

– Finally, the theoretical framing of the paper seems a bit unclear to me. The authors initially frame prEMG as an "alternative to SSRT" and "a more direct measure of the stopping latency" (p. 3f). However, they then argue that prEMG and SSRT are "partially independent" (which, incidentally, could be better phrased – measures are either independent or they are not). In the Conclusions, the authors then unequivocally state that "prEMG is a unique and reliable measure of stopping". So, if I understand the take home message correctly, the authors suggest that prEMG is a measure of stopping, which is (partially?) independent from SSRT, which, however, is also a "behavioral measure" of stopping (p. 23). In other words, the proposed relationship between prEMG, SSRT, and stopping should be more explicit. Do they both measure stopping? If so, how is that sensible? Or does only one of them? Should we discard SSRT in favor of prEMG? If the latter is to be the argument, I would put forward the above concerns, and reiterate that an explicit proof that prEMG reflects an inhibitory effort is still outstanding. If both SSRT and prEMG measure stopping, then how can they be only very moderately correlated and occur on different time scales?In summary, I remain rather unconvinced that prEMG is as of yet established as an "unambiguous marker of the stopping latency", which is the study's primary claim. It is also not clear to me if the additional set of problems associated with this measure makes it a viable "alternative to SSRT", as is implied. Hence, while I appreciate that two flawed measures (SSRT and prEMG) may be superior to only one, even if their relationship is unclear, I still believe it's too early to "encourage the widespread us of prEMG to investigate the underlying mechanisms of response inhibition".

The reviewer rightly points to some inconsistencies in our wording that muffle the main message. We are conceptualizing prEMG as an independent measure of action stopping that is not bounded by the assumptions of the horse race model. However, we do acknowledge that previous research on inhibition heavily relied on the SSRT and the horse race model, and we are thus interested in comparing prEMG against the existing framework. Further, prEMG and the SSRT share some variance, and thus our unfortunate wording that they are only ‘partially independent’. Guided by the reviewer’s helpful questions, we have now changed the discussion and conclusion of the manuscript considerably, emphasizing the points brought up by the reviewer particularly in the paragraphs entitled ‘Comparison of the prEMG peak latency and the SSRT’ and ‘Conclusions’.

Reviewer #2:Raud et al. explore the possibility of using partial electromyography responses on successful stop trials as a physiological measure to capture individual stopping latencies (prEMG). They provide an extensive overview of the measure in a large stop signal task data set, how it compares to traditional measures, how it relates to the assumptions underlying the horse-race model and its variability / reliability. In addition, it provides practical tips on how to collect the EMG data and to extract the partial responses. Overall, the data analysis is rigorous, and the conclusions are justified by the data. The main limitation of the approach is that partial responses were only detected on 23% of all successful stop trials. Previous literature has reported incidence of ~50%, but it remains a question whether this portion of the data is representative of all stopping behavior. The conclusion that the prEMG does not support the independent horse race model could have severe implications. It is not the first account of violations of the independence assumption of the model, which is widely used for the estimation of inhibition speed. It is vital to investigate the degree of violation in both response times and partial EMG responses under different task conditions to get an idea of the degree of the violation and its impact on the stopping latency estimations. I commend the efforts of the authors and would very much like the field to adopt the inclusion of these EMG recordings.1. The percentage of successful stops with a partial EMG response seems to be quite low (~23%). In my own experience the percentage was more around 40 (with a choice reaction time version of the SST) to 50% (with an anticipated response version of the SST) (data associated with Leunissen et al. 2017 Eur J Neuroscience, unpublished). Also in the paper of Coxon et al. 2006 and Jana et al. 2020 the percentage seems to be quite a bit higher (for the latter even 67%). I wonder whether the focus on the go task (and not slowing down here) can explain these differences. The RTs reported in your experiment are fairly long. Providing more feedback on the go task might help to increase the percentage of successful stops with a partial response. This would be important for at least two reasons: it is unclear how well 23% of all successful stops represent stopping behavior. Moreover, most researchers do not present 216 stop trials, but rather only 80-100. Leaving one with only ~10 partial responses per participant. The willingness to adopt such a lengthy protocol in addition to the EMG might be low, therefore I think it would be good to reflect on this.

We agree that a higher number of prEMG trials would have been preferable. We assume that the low number of prEMG trials was caused by a response device buttons with very low resistance. However, specifics in the task design that caused long goRTs may indeed also have contributed to the low number of prEMG trials.

We therefore investigated the association between the prEMG frequency and goRTs across the participants in our sample, as well as across different studies by a meta-analysis (lines 162-194; 706-723; Figure 2). Indeed, we found a negative relationship between the goRTs and the prEMG frequency, suggesting that shorter goRTs are associated with more prEMG trials (as hypothesized by the reviewer). This analysis suggests that specific task designs may help to increase the prEMG frequency (added in discussion lines 488-497). It would be particularly interesting to see whether the prEMG peak latency is in a similar range in tasks with less slowing, like the anticipated response task. While we know several studies that have analyzed the prEMG in this task, we are not aware that any of them have reported the prEMG latencies relative to the stop signal. Of note, most of the studies used in the meta-analysis had less stop trials available than in the current study, yet produced similar estimates of the prEMG peak latencies. Thus, lengthy protocols do not seem to be necessary.

2. APB vs FDI: You report the partial EMG responses from a downwards thumb movement in the APB. Of course, this makes sense for this movement direction, but in my experience it is much easier to get a good signal to noise ratio in the EMG for the FDI (which would require and abduction of the index finger or a different hand position). See Coxon et al. 2006 J Neurophysiology Figure 1 and 3 for a comparison of the EMG traces of the APB and FDI. Having a better SNR might also help in identifying the partial responses in a greater number of trials.

The meta-analysis indicates that about half of the prEMG-latency studies report EMG from the APB, while the rest reports from the FDI (or a combination of both). We have experienced that the signal to noise ratio is highly dependent on the movement direction and electrode placement. After some experimenting with both index finger (FDI) and thumbs (APB), we found better signal-to-noise ratio for thumbs in our setting, but we (and others) have also achieved good signal from FDI with different set-up and response device. We have now elaborated on the choice of the recorded muscle in Box 3.

3. You show a dependency between the go and stop process based on the EMG analyses. It might be of added value to check the level of dependency in the response times as well. For example based on the BEEST approach outlined by Dora Matzke in this pre-print (https://psyarxiv.com/9h3v7/). It would be interesting to see if there is agreement between the two approached with respect to the (lack) of independence. It would also be nice to be able to compare the RTs on unsuccessful stops and the fast go-RT portion (discussed on page 14, line 215 but not shown).The conclusion that there is dependency between the go and stop process seems to hinge on the finding that the average EMG onset in unsuccessful stop trials was slower than the average EMG onset in fast go trials. Could a difference in power perhaps produce this result? You have many more go trials than unsuccessful stop trials, would the full distribution simply not have been sampled in the unsuccessful stop trials?

Thank you for the reference to the procedure proposed by Matzke et. al., as this is an interesting addition to the context independence analysis. We ran the suggested analysis for the RT data and generalized it to the EMG data. (Note that for comparison with the EMG data, our analysis is based on the observed data instead of predictions based on BEESTS posterior parameters). The cumulative density functions (CDFs) and corresponding delta plots are visualized in Author response image 1. For the RT data, we found the expected fanning effect. This was also the case for EMG onsets comparing the unsuccessful stop trials and go trials. When we calculated CDFs for all stop trials with EMG combined (unsuccessful + successful trials with prEMG), this CDF was also leftwards from the go-CDF, showing a fanning effect at higher quantiles. When the CDFs were calculated separately for the successful stop trials with prEMG, the go trial CDF dominated the prEMG CDF in the lower quantiles, indicating distribution deviance between go and successful stop trials with prEMG at fast responses. Yet, this may be expected, as the stop trials with fast EMG onsets end up being unsuccessful stop trials, thus the prEMG distribution appears to be censored both at the left and right side.

**Author response image 1. sa2fig1:** Cumulative density functions (CDFs) and delta function for reaction time (RT; upper panels) and EMG data (lower panels). X-axis (t) in CDF plots represents normalized response times, and the y-axis (F(t)) represents probability of obtaining a response time faster or equal to t. The delta functions represent the horizontal difference between the go and stop trial CDFs as a function of cumulative probability. The combined stop variables include unsuccessful and successful stop trials with prEMG. The shaded areas represent 95% confidence intervals across participants. Note that successful stop (prEMG) CDF (lower left panel) is likely inaccurate at high t’s, due to very few available trials with slow EMG onsets.

While we find this analysis very interesting, we decided against including it to the manuscript. This is because the theoretical predictions about the prEMG are not well worked out in this framework. Further, according to Matzke et. al., the observed CDFs may only detect severe violations compared with the BEESTS-based CDFs, but no modelling approach exists for the prEMG. Lastly, to explain this analysis sufficiently would lengthen the already dense results section considerably, whereas the results do not add much new information, as the results that are currently included already point to a similar conclusion - trials with prEMG seem to be those with medium response times. Critically, the context dependence test we had already included is more stringent than the one based on the CDFs, as it directly compares the equivalence of the corresponding observable proportions of the go and stop trial distributions. We have now included this test also for the RTs, as suggested by the reviewer (lines 254-259).

To assess whether differences in the sampling of EMG in unsuccessful stop and go trials could drive the reported EMG onset difference, we ran a control analysis. First, for each participant, we randomly drew a matching number of fast go and medium go trials (on average 105 fast trials, corresponding to the average number of unsuccessful stop trials, and 23 medium trials, corresponding to the average number of prEMG trials). Second, we did two paired t-tests: comparing the sub-sample of fast go trials with the unsuccessful stop trials, and medium go trials with the stop trials with prEMG. Third, we repeated this procedure 1000 times to obtain t- and p-value distribution. As seen in Author response image 2, the t-values are always negative for the unsuccessful stop trials and positive for the prEMG trials, corresponding with the pattern reported in the main manuscript using all values. In addition, the p-values are below 0.05 every single time. In conclusion, we are confident that the results are not affected by the differences in sampling. This analysis is now references in lines 276-280.

4. In my opinion the discussion could state a bit more clearly what the benefit of using the EMG peak latency on successful stop trials is compared to a parametric approach of analyzing the response times for example. Should it really be EMG or could one also simply use a force censor to obtain a similar measure? What are possible disadvantages of EMG? For example, in clinical populations one will find that the background EMG is much higher, making it more difficult to identify the partial responses.

This is an excellent point. We do think that other parametric approaches are viable options as well, but we chose not to dwell on this point not to diverge from the main message about EMG properties. We have now indicated that alternative parametric approaches, such as force sensors, may outperform EMG in some contexts (e.g., in certain clinical populations), if properly validated (lines 493-497).

5. I think it would be helpful to include a visualization of the stop signal task used.6. Please include a figure that visualizes how the different latencies are determined.

These are now depicted in Figure 1 and Box 2 for point 5 and 6, respectively.

7. Rise time is defined as EMG onset to peak latency. But peak latency was the time between the stop signal and the peak amplitude in the EMG, obviously there is no peak latency in go trials. Please redefine for clarity.

We have clarified that for the rise time calculation, the onset and peak latency are extracted relative to the same signal, i.e. go signal (line 143 and Table 2).

8. Motor time was longer in unsuccessful stop trials than in go trials. Was this true for all portions of go trials in the distribution? Response force on go trials tends to increase with the increasing likely hood of a stop signal appearing (i.e. participants wait and then respond very quickly and forcefully) (van den Wildenberg et al. 2003 Acta Psychol). Those trials might have a higher RT but a very short motor time.

That is an interesting observation and merits further investigation. The appropriate analysis would then be to test the equivalence of motor times in fast go trials and unsuccessful stop trials. We calculated the motor times for the three parts of the go-RT distribution, as outlined in the section of contextual independence testing. In contrast to the reviewer’s prediction, we found increasing motor times with increasing RTs, and that slow go trial motor times roughly correspond to unsuccessful stop motor times. We added this analysis to the manuscript (lines 145-161), but avoided too much speculation on the implications, as the replication analysis on a different dataset showed slight deviation from the overall pattern.

9. Could the average areas under the curve be added to table 2? This would be helpful for understanding the analysis on the different dynamics of going and stopping.

These have now been added to the table 2.

10. Page 24, line 370-371. Comment regarding SSRT being associated with a speed-accuracy trade off suggests that this is (possibly) not the case for prEMG. Give that this speed-accuracy tradeoff reflects the strategic choices of the participant I assume that this should also be reflected in the (number, size, ...) of the prEMG.

Yes, there is evidence for some sort of trade-off between going and stopping. Previously, we mentioned trade-off between fast going and accurate stopping based on previous literature. Here, we do not actually look at stopping accuracies, but do find evidence for a trade-off between fast going and slow stopping, or vice versa. Interestingly, this appears to strongly affect the SSRTs and prEMG detection frequency, but to a lesser degree prEMG peak latency. The correlation analysis is reported on lines 171-177, 184-187, and we have briefly included it in the discussion lines 456-462.

References

Atsma J, Maij F, Gu C, Medendorp WP, Corneil BD. 2018. Active Braking of Whole-Arm Reaching Movements Provides Single-Trial Neuromuscular Measures of Movement Cancellation. *J Neurosci* 38:4367–4382. doi:10.1523/JNEUROSCI.1745-17.2018

Coxon JP, Stinear CM, Byblow WD. 2006. Intracortical inhibition during volitional inhibition of prepared action. *J Neurophysiol* 95:3371–3383. doi:10.1152/jn.01334.2005

Goonetilleke SC, Doherty TJ, Corneil BD. 2010. A within-trial measure of the stop signal reaction time in a head-unrestrained oculomotor countermanding task. *J Neurophysiol* 104:3677–3690. doi:10.1152/jn.00495.2010

Goonetilleke SC, Wong JP, Corneil BD. 2012. Validation of a within-trial measure of the oculomotor stop process. *Journal of Neurophysiology* 108:760–770. doi:10.1152/jn.00174.2012Havas JD, Ito S, Gomi H. 2020. On Stopping Voluntary Muscle Relaxations and Contractions: Evidence for Shared Control Mechanisms and Muscle State-Specific Active Breaking. *J Neurosci* 40:6035–6048. doi:10.1523/JNEUROSCI.0002-20.2020

Tatz JR, Soh C, Wessel JR. 2021. Common and Unique Inhibitory Control Signatures of Action-Stopping and Attentional Capture Suggest That Actions Are Stopped in Two Stages. *J Neurosci* 41:8826–8838. doi:10.1523/JNEUROSCI.1105-21.2021